# mTOR-dependent translation drives tumor infiltrating CD8+ effector and CD4+ Treg cells expansion

**Benedetta De Ponte Conti[1,2], Annarita Miluzio[3], Fabio Grassi[1,4], Sergio Abrignani[3,5], Stefano Biffo[3,6]\*, Sara Ricciardi[3,6]\***

[1]Institute for Research in Biomedicine, Università della Svizzera Italiana (USI), Bellinzona, Switzerland; [2]Graduate School of Cellular and Molecular Sciences, University of Bern, Bern, Switzerland; [3]Istituto Nazionale Genetica Molecolare "Romeo ed Enrica Invernizzi", Milan, Italy; [4]Department of Medical Biotechnology and Translational Medicine, Universita` degli Studi di Milano, Milan, Italy; [5]Department of Clinical Sciences and Community Health, Università degli Studi di Milano, Milan, Italy; [6]Bioscience Department, Università degli Studi di Milano, Milan, Italy

**\*For correspondence:**
biffo@ingm.org (SB);
ricciardi@ingm.org (SR)

**Competing interest:** The authors declare that no competing interests exist.

**Abstract** We performed a systematic analysis of the translation rate of tumor-infiltrating lymphocytes (TILs) and the microenvironment inputs affecting it, both in humans and in mice. Measurement of puromycin incorporation, a proxy of protein synthesis, revealed an increase of translating CD4+ and CD8+ cells in tumors, compared to normal tissues. High translation levels are associated with phospho-S6 labeling downstream of mTORC1 activation, whereas low levels correlate with hypoxic areas, in agreement with data showing that T cell receptor stimulation and hypoxia act as translation stimulators and inhibitors, respectively. Additional analyses revealed the specific phenotype of translating TILs. CD8+ translating cells have enriched expression of IFN-γ and CD-39, and reduced SLAMF6, pointing to a cytotoxic phenotype. CD4+ translating cells are mostly regulatory T cells (Tregs) with enriched levels of CTLA-4 and Ki67, suggesting an expanding immunosuppressive phenotype. In conclusion, the majority of translationally active TILs is represented by cytotoxic CD8+ and suppressive CD4+ Tregs, implying that other subsets may be largely composed by inactive bystanders.

## Editor's evaluation

By employing melanoma and colon cancer allograft mouse models, the authors show that tumor-infiltrating T-cells exhibit heterogeneity in levels of protein synthesis that correlates with their immunophenotypes. Moreover, some evidence is provided that the observed heterogeneity in protein synthesis rates in tumor-infiltrating T-cells levels is driven by distinct conditions in different parts of tumor microenvironment. Overall, these findings further corroborate the importance of mRNA translation in immune cell plasticity and suggest that relying on monitoring steady-state mRNA levels may not provide the full picture of immunophenotypes in tumor niche.

## Introduction

Cancer immunoediting is the dynamic process whereby cancer cells modify the immune microenvironment to their advantage. In a typical cancer scenario, altered protein products caused by genetic mutations function as neoantigens, eliciting an immune response against cancer cells (*Schumacher*

*and Schreiber, 2015*). In parallel, the inflammatory, hypoxic, and often necrotic microenvironment of tumors modifies the immune response (*Kalluri, 2016*). Tumor-infiltrating T lymphocytes (TILs) are the major players in the immune response (*Quail and Joyce, 2013*). Conventional classification of T lymphocytes divides them in CD8$^+$ cytotoxic T cells, and various subtypes of CD4$^+$ cells, each with a precise range of action. In general, CD8$^+$ cytotoxic T cells, CD4$^+$ T-helper 1 cells producing interferon-γ, and natural killer cells are associated with favorable anti-tumor immune responses (*Tosolini et al., 2011*). In contrast, myeloid-derived suppressor cells, and CD4$^+$ FOXP3$^+$ regulatory T (Treg) cells producing IL-10 and TGF-β have immunosuppressive and protumoral effects (*Shang et al., 2015*). Other T cell subsets, present at lower frequency, have been linked to patient survival in specific tumors, such as CD4$^+$ Tr1 (*Bonnal et al., 2021*), or CD4$^-$ CD8$^-$ unconventional αβ T cells (*Ponzetta et al., 2019*). Recently, single-cell RNAseq studies have extended the phenotypic description of TILs (*De Simone et al., 2016*; *Guo et al., 2018*; *Miller et al., 2019*; *Stuart and Satija, 2019*; *Yao et al., 2019*; *Zheng et al., 2017*) showing distinct patterns of immune activation and exhaustion, and increasing the number of T cell subtypes. Although many of these subtypes are present across different tumor types, not all cancer types with similar TIL landscapes respond similarly to immunotherapy (*Paijens et al., 2021*). These observations highlight the complexity of the tumor-immune interactions and suggest the presence of environmental modifiers of the T cell transcriptional repertoire. Notably, in spite of the elevated number of T cell players found in the tumor microenvironment, the most consistent survival predictors are (i) the number of TILs and (ii) the ratio between CD8$^+$ cytotoxic T cells and CD4$^+$ FOXP3$^+$ Tregs (*Gooden et al., 2011*).

The high complexity of the T cell transcriptional repertoire is only one side of the coin. Several studies show that the presence of a messenger RNA (mRNA) may not reflect the active synthesis of the encoded protein (*Schwanhäusser et al., 2011*). It is known that some mRNAs are translated poorly and only in response to specific stimuli (*Pelletier and Sonenberg, 2019*). Regulation of protein synthesis, called translational control, occurs both at the global and at the specific mRNAs level. Protein synthesis is one of the most energy-consuming pathways in cell metabolism (*Buttgereit and Brand, 1995*), and microenvironment conditions and nutrients are major controllers of the global rate of synthesis (*Biffo et al., 2018*). For instance, hypoxia leads to the downregulation of high-energy processes, including protein synthesis (*Horman et al., 2002*; *Liu et al., 2006*). Local conditions of amino acid concentration affect translation, and global translation decreases in response to amino acid deprivation. Both hypoxia and amino acid deprivation activate also specific translation. When an essential amino acid is limiting, the concentration(s) of the non-aminoacylated ('uncharged') tRNA increase(s). Uncharged tRNAs bind to the regulatory domain of GCN2 kinase causing its activation, the phosphorylation of eIF2α, a shutdown of global protein synthesis, and the specific derepression of the translation of uORF-controlled mRNAs (*Wek, 2018*). Four kinases (eIF2AK1–4) are able to phosphorylate eIF2α (*Loreni et al., 2014*) and are all expressed in T cells (*Mitchell et al., 2015*). Hypoxia is a stimulator of eIF2AK3 known also as PERK (*Koumenis et al., 2002*). Consequently, the hypoxic tumor microenvironment may inhibit protein synthesis through specific cascades, leading to a mismatch between the presence of mRNAs and their encoded proteins. It is therefore evident that translational inhibition may be particularly relevant for T cells that respond to the conditions of the microenvironment (*Biffo et al., 2018*).

Stimulation of protein synthesis is one of the hallmarks of T cell exit from quiescence, beginning of growth and proliferation and is associated with both quantitative (*Garcia-Sanz et al., 1998*) and qualitative changes (*Mikulits et al., 2000*) in translation. Stimulation of protein synthesis occurs through the nutrient growth factor receptor cascades that converge on initiation factors eIF4E, through PI3K-mTORC1, ERK-MNK, and eIF6 (*Loreni et al., 2014*; *Bhat et al., 2015*). Activation of the T cell receptor leads to a robust activation of the growth factor cascade. In particular, stimulation of the mTORC1-S6K branch is of outmost importance for T cell function. mTORC1 is a major controller of both translation and T cell biology (*Bjur et al., 2013*; *Piccirillo et al., 2014*). Inhibition of mTORC1 by rapamycin has profound effects on T cell response and polarization (*Powell and Delgoffe, 2010*). The relevance of the translational branch in driving mTORC1 responses is demonstrated by the fact that ablation of 4EBP1 and 4EBP2, which are translational modulators phosphorylated by mTORC1, rescues the inhibitory action on proliferation of raptor depletion (*Dowling et al., 2010*). Several mRNAs are rapidly engaged following T cell stimulation, promoting an immediate translational and glycolytic switch to ramp up the T cell activation program (*Wolf et al., 2020*). Recently, we found that stimulation of

fatty acid synthesis in T cells requires the translational activation of the rate-limiting enzyme ACC1. Intriguingly, ACC1 mRNA was expressed but not translated also in quiescent T cells (*Ricciardi et al., 2018*). Translational regulation of glycolytic enzymes was seen also in activated CD4[+] cells (*Manfrini et al., 2020b*), in line with other experimental models. Another translation factor, eIF5A, promotes the expression of a subset of mitochondrial proteins involved in the TCA cycle and oxidative phosphorylation (*Puleston et al., 2019*). In short, in vitro, T cells rapidly respond to T cell receptor stimulation by increasing protein synthesis.

The presence of a large layer of translational control in T cells raises the question whether TILs are efficiently translating in vivo, which environmental stimuli regulate their translation, and whether active translation affects their developmental pathway. In this study we demonstrate that, in vivo, suppressive CD4[+] Tregs and cytotoxic CD8[+] T cells are the major populations of translating lymphocytes. Hypoxia and mTOR signaling are major modulators of lymphocyte translation, and translating lymphocytes are phenotypically different from translationally inactive lymphocytes. We hypothesize that the predictive value on survival of suppressive CD4[+] Tregs and cytotoxic CD8[+] T cells is due to their capability to be translationally active amid a large percentage of not translating lymphocytes.

## Results

### Translational efficiency is increased by TCR signaling and inhibited by hypoxia

Puromycin, by entering the acceptor site of ribosomes and incorporating into nascent polypeptide chains, represents a valid tool to quantify protein synthesis within cells in vivo. We used puromycin measurement in combination with quantitative immunoblotting to capture global protein synthesis rate in human primary lymphocytes (*Figure 1A*). We have previously shown that naïve T cells have a poised translational machinery, with pre-accumulated mRNAs that are efficiently translated only after T cell receptor stimulation (*Ricciardi et al., 2018*). This prompted us to further evaluate the relationship between T cell receptor stimulation and translation. T cell receptor stimulation leads to a progressive, temporal, steady increase of protein synthesis (*Figure 1B*). Next, we checked whether the strength of T cell receptor stimulation affected the global output of proteins. To this end, primary lymphocytes were briefly cultivated in the presence of two different amounts of anti-CD3/CD28, and protein synthesis was measured by puromycin incorporation (*Figure 1C*). We saw higher puromycin incorporation in cells stimulated with the higher concentration of anti-CD3/CD28 (*Figure 1C*). Phosphorylation of ribosomal protein rpS6 occurs in serine 235/236 by S6K1-2 downstream of mTORC1 and by RSKs kinases downstream of ERK, as well as in serine 240/244 by S6K1-2 kinases (*Meyuhas, 2015*). All these pathways are activated by TCR stimulation (*Piccirillo et al., 2014*). Consistently, rpS6 phosphorylation increased from undetectable to strong levels after anti-CD3/CD28 stimulation (*Figure 1C*). rpS6 itself was detectable before TCR stimulation and increased upon stimulation, in line with previous data (*Wolf et al., 2020*; *Araki et al., 2017*).

Activation of protein synthesis correlates with rpS6 phosphorylation (*Figure 1C*). Inhibition of mTOR pathway by the mTOR inhibitor PP242 leads to both the reduction of rpS6 phosphorylation and a conspicuous inhibition of puromycin incorporation, that is, protein synthesis (*Figure 1D*). The ERK pathway converges on Mnk1/2 that phosphorylate Ser209 of eIF4E (*Roux and Topisirovic, 2018*). In vitro, Mnk inhibition did not consistently reduce puromycin incorporation, indicating that this pathway is not massively involved in global translation of T cells (*Figure 1D*).

Since infiltrating lymphocytes may encounter an environment rich in hypoxic areas, we wondered if hypoxia was able to rapidly modify the capability of lymphocytes to translate in response to T cell receptor stimulation. We therefore quantitated the rate of puromycin incorporation in the presence of hypoxic environment (*Figure 1E*). Hypoxia resulted in a rapid reduction of the translation rate of primary lymphocytes (*Figure 1F*, *Figure 1—source data 3*). Several translation pathways are modulated by hypoxia, including the one driven by PERK. PERK phosphorylates Ser51 of eIF2α (*Koumenis et al., 2002*). We did not detect eIF2α phosphorylation by Western blotting, likely due to the limited number of human cells. We used a modified ELISA procedure and found that hypoxia induced eIF2α phosphorylation (*Figure 1G*, *Figure 1—source data 4*). On the basis of these data, we hypothesize that, in vivo, the combination of T cell receptor stimulation and hypoxic conditions is a major determinant of the translational capability of lymphocytes.

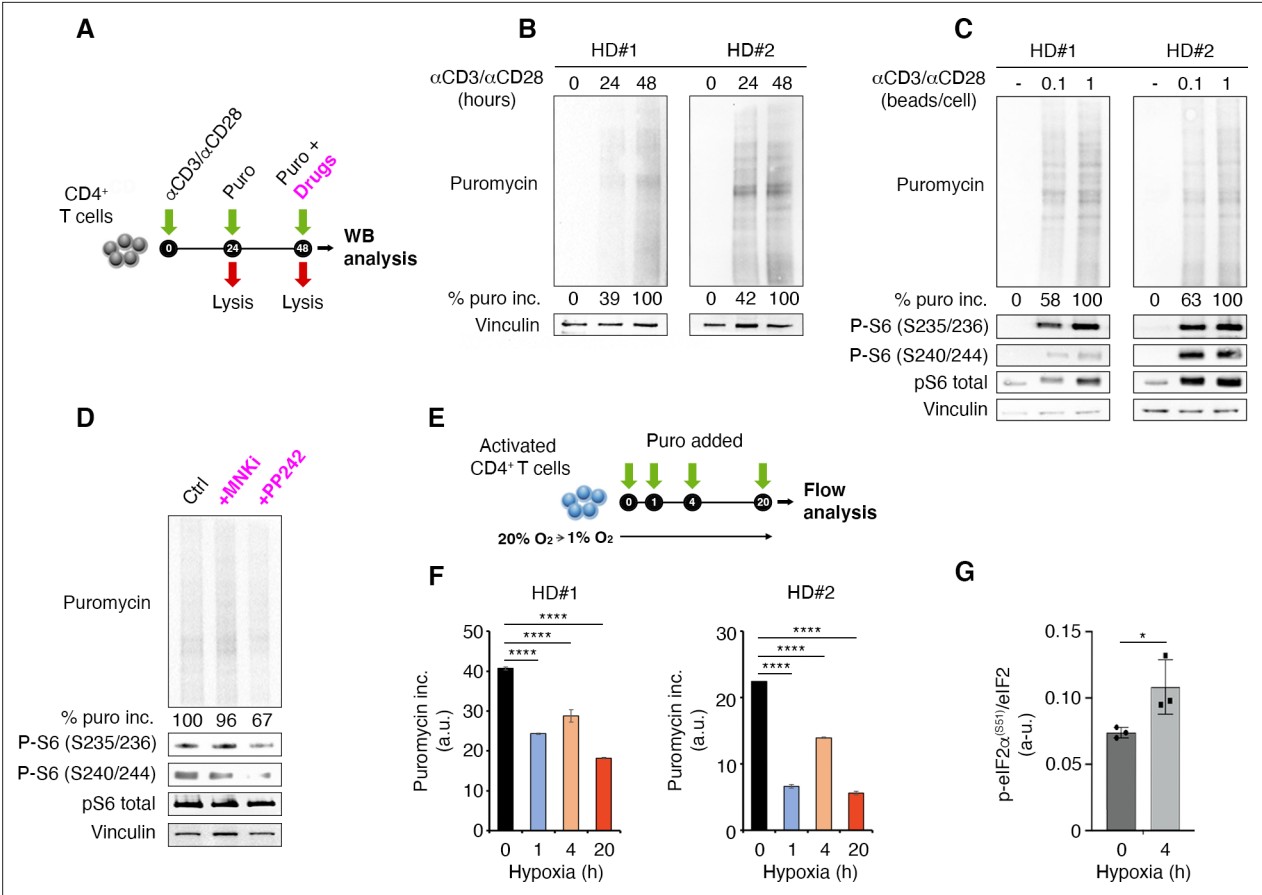

**Figure 1.** T cell receptor stimulation through mTORC1 is a strong stimulator of translation counterbalanced by hypoxia in primary lymphocytes. (**A**) Schematic diagram for the experiment. (**B**) Immunoblot of puromycin incorporation in CD4+ T cells following stimulation by anti-CD3/CD28 at the indicated time points shows that T cell receptor stimulation leads to a progressive increase of puromycin in cultured lymphocytes. The immunoblot shows two replicates with cells isolated from one healthy donor per experiment. Densitometry normalized to vinculin. (**C**) The strength of T cell receptor stimulation correlates with puromycin incorporation. The immunoblot shows two replicates with cells isolated from one healthy donor per experiment. (**D**) mTOR inhibition reduces rpS6 phosphorylation and translation. Stimulated CD4+ T cells were treated for 30 min with either 2 µM PP242 or 3 µM MNK inhibitor before collecting extracts. Puromycin incorporation and phosphorylation of rpS6 were measured by Western blotting. The immunoblot is representative of the pool of two replicates with cells isolated from one healthy donor per experiment. (**E and F**) Hypoxic environment sharply reduces translation. Stimulated CD4+ T cells were transferred from 20 % $O_2$ to 1 % $O_2$ for the indicated times (**E**), and translation was measured by flow cytometry (**F**). Data are mean ± s.d. p Values are determined by ANOVA with Dunnett's post hoc test. ****$p < 0.0001$. (**G**) Hypoxia induces phosphorylation of eIF2α. Stimulated CD4+ T cells were incubated for 4 hr under normoxia or hypoxia and the phosphorylation of eIF2α was determined by ELISA assay as described in Materials and methods using an anti-phospho-eIF2α-specific antibody. The data represents the pool of three independent experiments. Data are mean ± s.d. p Values are determined by two-tailed Student's t-tests. ****$p < 0.001$.

The online version of this article includes the following figure supplement(s) for figure 1:

**Source data 1.** Source data for *Figure 1B, C, D, F, and G*.

**Source data 2.** Source data for *Figure 1B, C, D, F, and G*.

**Source data 3.** Source data for *Figure 1B, C, D, F, and G*.

**Source data 4.** Source data for *Figure 1B, C, D, F and G*.

## Only a minority of TILs actively translates and is characterized by an mTORC1 activated phenotype

Next, we addressed the question of whether, in vivo, we could observe heterogeneity in the translational efficiency of TILs. To answer this question, we engrafted subcutaneously C57BL/6 mice with B16F10 melanoma cells (*Figure 2A*). We administered a single injection of puromycin (50 mg/kg body mass) and after 1 hr, mice were sacrificed and T cells isolated for analysis in flow cytometry (*Figure 2A*). Then, the amount of protein synthesized was quantified on the basis of puromycin incorporation by

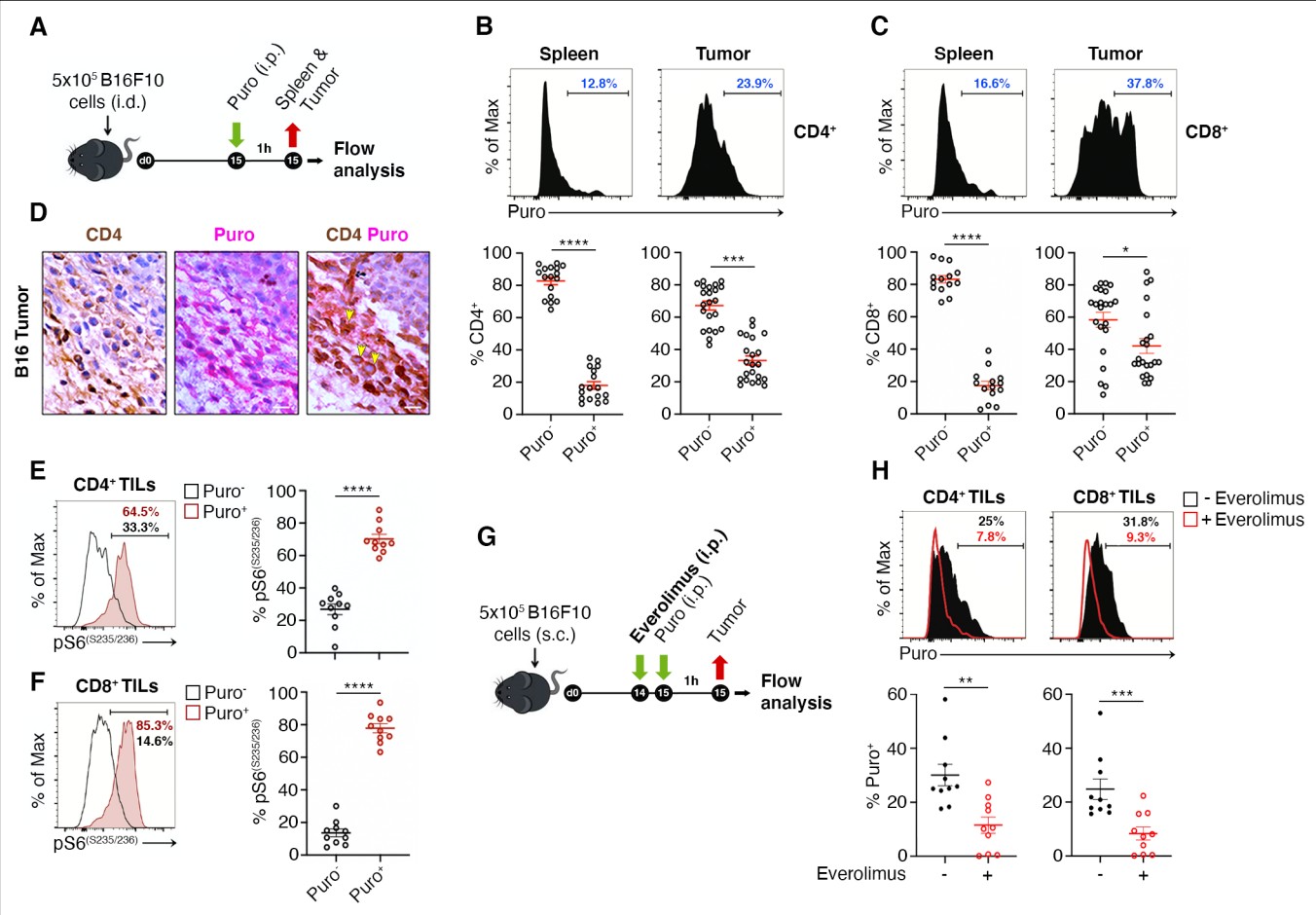

**Figure 2.** The tumor microenvironment stimulates translation in a limited number of tumor-infiltrating lymphocytes (TILs) with activated mTORC1 pathway. (**A**) Experimental outline. Tumor cells were injected in recipient mice. Puromycin was injected intraperitoneally, and T cells were collected 1 hr later for the analysis. (**B and C**) The amount of incorporated puromycin was determined by FACS analysis in CD4+ (**B**) or CD8+ lymphocytes (**C**). Representative plots and statistical analysis (mean ± SEM) show that the number of Puro+ TILs is always higher in tumors versus spleen and that only parts of TILs are translationally active. Percentages of positive cells in each gate are shown. Data from three experiments pooled together (n = 5–8 mice per experiment) *p < 0.05; ***p < 0.001; ****p < 0.0001. (**D**) Immunohistochemical analysis of puromycin in CD4+ TILs. Puro+ cells (pink) are concentrated in some tumor areas, where clusters of highly translating CD4+ cells (brown) are found. Scale bars, 10 μm. (**E**) Representative plots (left) and statistical analysis (mean ± SEM) for pS6(S235/236) within Puro+ and Puro- CD4+ and CD8+ TILs shows that, in vivo, most translating TILs have the mTORC1-S6K pathway active. Percentages of positive cells in each gate are shown. Data from two experiments pooled together (n = 5 mice per experiment) ****p < 0.0001. (**G**) Experimental outline. Tumor cells were injected in recipient mice that were given intraperitoneal injections of 5 mg/kg everolimus for 2 consecutive days. Puromycin was injected 1 hr before sacrificing mice and collecting TILs cells for subsequent flow cytometry analysis. (**H**) Representative flow cytometry plots and statistical analysis (mean ± SEM) for puromycin within CD4+ and CD8+ TILs showing that the mTOR inhibitor everolimus reduces translation. Percentages of positive cells in each gate are shown. Data from two experiments pooled together (n = 10 mice per experiment) **p < 0.01; ***p < 0.001.

The online version of this article includes the following figure supplement(s) for figure 2:

**Source data 1.** Source data for *Figure 2B, C, E, F and H*.

**Figure supplement 1.** Flow cytometry gating strategy analysis of puromycin incorporation in CD4+ and CD8+ T cells, either in the spleen or in tumor.

**Figure supplement 2.** The tumor microenvironment stimulates translation in a limited number of tumor infiltrating lymphocytes (TILs) with activated mTORC1 pathway.

**Figure supplement 2—source data 1.** Source data for *Figure 2—figure supplement 2B-E*.

FACS. The same gating strategy was applied for spleen lymphocytes and TILs (*Figure 2—figure supplement 1*). Intriguingly, we found that at the single-cell level, in vivo, the translational efficiency of both CD4+ and CD8+ TILs was highly variable (*Figure 2B and C*, *Figure 2—source data 1*), namely some TILs had no puromycin incorporation, others were highly positive for puromycin. On the basis of

the gating shown in *Figure 2B and C*, we defined two categories of infiltrating T cells, high puromycin (Puro$^+$) and low puromycin (Puro$^-$) T cells, reflecting translating T cells and translationally silent T cells, respectively. This simple analysis unveiled that CD4$^+$ TILs translated more than spleen-resident CD4$^+$ T cells (*Figure 2B*). The number of highly translating CD4$^+$ TILs in different animals ranged between 20% and 60% (*Figure 2B*). Similar to CD4$^+$ TILs, the number of highly translating CD8$^+$ TILs was more than twofold higher than in the spleen (*Figure 2C*). Within individual tumors, Puro$^+$ CD8$^+$ TILs ranged between 20% and 90% (*Figure 2C*). Taken together these data demonstrate the presence of a strong layer of translational control, driven by the microenvironment, that affects the biological activity of individual lymphocytes.

IHC on paraffin-embedded B16 tumor sections also confirmed that (i) the levels of incorporated puromycin were heterogenous inside the tumor, with areas having high expression mixed with areas devoid of expression (*Figure 2D*), and that (ii) scattered lymphocytes show a robust puromycin immunoreactivity (*Figure 2D*). To substantiate these results within a different tumor microenvironment, we engrafted C57BL/6 mice with MC38 colon adenocarcinoma cells. Similar to what was obtained for B16 TILs, we found that TILs have significant higher levels of puromycin signal over spleen-resident T cells, and that, in vivo, within the tumor, only a fraction of lymphocytes is translationally active (*Figure 2— figure supplement 2A*).

We then addressed whether puromycin incorporation, in vivo, was associated with mTORC1 signaling. TCR stimulation causes an increase of mTORC1 activity that can be detected by the analysis of rpS6 phosphorylation (*Ricciardi et al., 2018*). Notably, both CD4$^+$ (*Figure 2E*, *Figure 2—source data 1*) and CD8$^+$ (*Figure 2F*, *Figure 2—source data 1*) cells containing high levels of puromycin were consistently positive for rpS6 phosphorylation. We found that phosphorylation of rpS6 in the puromycin-positive cells was significantly higher both at Ser235/236 (*Figure 2E–F*) and at Ser240/244 (*Figure 2—figure supplement 2B-C*, *Figure 2—figure supplement 2—source data 1*). In order to verify that also in vivo, as in vitro (*Figure 1*), inhibition of the mTORC1 pathway affected the translation of TILs, we treated mice with mTOR inhibitor everolimus (*Figure 2G*). In short, treatment of mice with everolimus caused a decrease in the percentage of puromycin-positive TILs (*Figure 2H*, *Figure 2—source data 1*), and a slight but significant decrease in rpS6 phosphorylation (*Figure 2— figure supplement 2D*, *Figure 2—figure supplement 2—source data 1*).

Mnk inhibition in vitro did not result in significant changes in puromycin incorporation (*Figure 1*). However, in vivo, a very small pool of TILs was positive for p-eIF4E, and clearly segregated with high puromycin incorporation (*Figure 2—figure supplement 2E*, (*Figure 2—figure supplement 2— source data 1*)). In conclusion, the activation of specific signaling pathways, at the single-cell level, explains high puromycin incorporation. Among them, the mTORC1 pathway affects the global translation rate of a substantial percentage of TILs.

## Hypoxic niches reduce puromycin incorporation

Next, the relationship between puromycin incorporation and hypoxia, in vivo, was analyzed. We injected mice with pimonidazole (PMO), a chemical probe that forms protein adducts in viable hypoxic cells (*Rademakers et al., 2011*) and puromycin (*Figure 3A*) and performed stainings on tissue sections for the endothelial marker CD31. It was found that the amount of tumor areas that were not vascularized was limited, the maximal distances between vessels being in the range of 150–200 µm (*Figure 3B*). Subsequently, we purified both CD4$^+$ (*Figure 3C*) and CD8$^+$ (*Figure 3D*) TILs and analyzed by FACS analysis the relationship between PMO and puromycin incorporation. In spite of the relative absence of highly hypoxic tumor areas, the data show that, in vivo, an inverse relationship between PMO staining and puromycin exists (*Figure 3C and D*, *Figure 3—source data 1*). Next, we performed immunofluorescence analysis. We clearly detected cells triple-stained for PMO, CD4$^+$ and p-eIF2α−like immunoreactivity in hypoxic areas (*Figure 3E*), but not in non-hypoxic areas (*Figure 3— figure supplement 1*). We suggest that hypoxia acts as a repressor of translation also in vivo, in TILs. These data raise the question whether the phenotype of TIL is specific.

## High translation rate in CD8$^+$ cells correlates with IFN-γ production and CD-44 expression

We verified the existence of a relationship between high translation and the CD8$^+$ phenotype by measuring through flow cytometry the expression of markers of memory and effector T cell

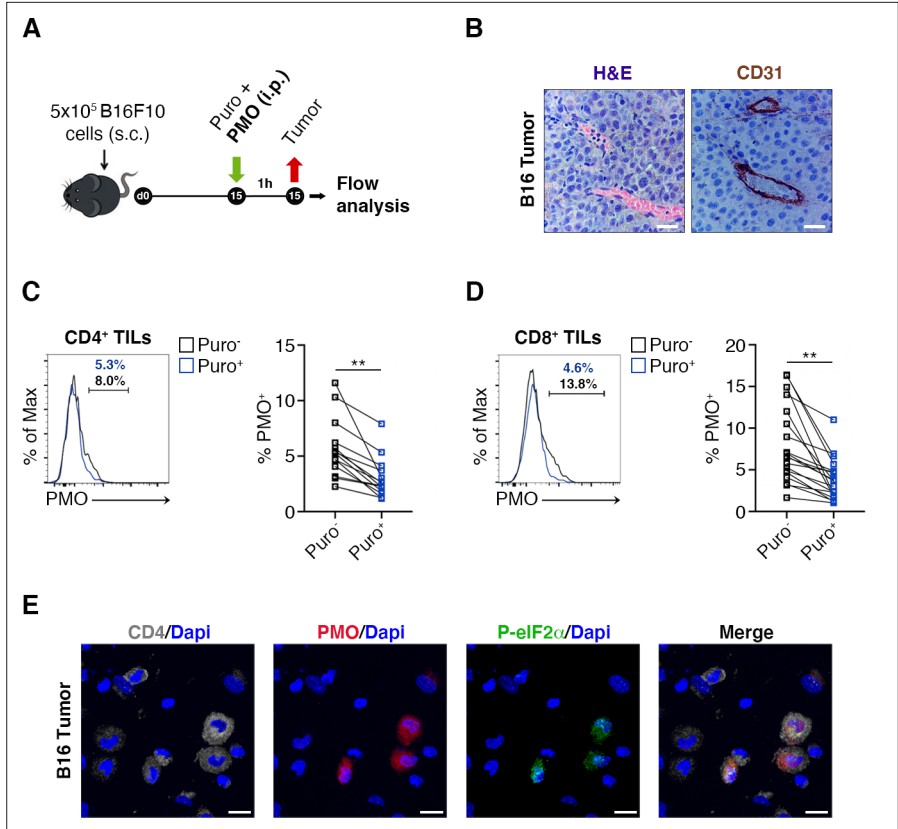

**Figure 3.** Hypoxia limits translation of tumor-infiltrating lymphocytes (TILs) in vivo. (**A**) Experimental outline. Tumor cells were injected in recipient mice. Puromycin was injected intraperitoneally together with the hypoxia marker pimonidazole (PMO). T cells were collected 1 hr later for the analysis. (**B**) CD31 staining in tumor specimens shows ample vascularization with limited areas far from blood vessels suggesting the absence of truly hypoxic areas and limited hypoxic gradients. Scale bars, 15 µm. (**C and D**) Representative plots (left) and statistical analysis (mean ± SEM) for PMO+ within Puro+ and Puro- CD4+ and CD8+ TILs show, in vivo, the inverse correlation between hypoxia and translational efficiency. Percentages of positive cells in each gate are shown. Data from two experiments pooled together (n = 7–8 mice per experiment) **p < 0.01. (**E**) Representative immunofluorescence images showing that in tumor specimens the majority of PMO+ CD4+ TILs are p-eIF2α(S51)+. Scale bars, 10 µm.

The online version of this article includes the following figure supplement(s) for figure 3:

**Source data 1.** Source data for *Figure 3C and D*.

**Figure supplement 1.** Immunofluorescence staining of CD4 (in grey), PMO (in red), and p-eIF2α (in green) in B16 spleen specimens.

differentiation, activation, as well as of relevant costimulatory and coinhibitory molecules. The experimental design is shown in *Figure 4A*. First of all, we observed a maximum of puromycin incorporation in CD44+ CD8+ T cells, indicating that Puro+ CD8+ TILs largely retain an activated-like phenotype (*Figure 4B*, *Figure 4—source data 1*). We co-stained freshly isolated CD8+ TILs for puromycin and the following sets of markers: (i) PD-1, TIM3, CTLA-4, and TIGIT that characterize CD8 T cell exhaustion within the tumor microenvironment (*Anderson et al., 2016*); (ii) the T cell receptor costimulatory proteins CD28, ICOS, SLAMF6, and CD27, pro-inflammatory IFN-γ and TNF-α cytokines, Granzyme B; (iii) tissue-resident T cells markers CD103, CD69, and CCR5 (*Golubovskaya and Wu, 2016*); (iv) ecto-nucleoside triphosphate diphosphohydrolase-1 CD39 (*Takenaka et al., 2019*). We decided to apply a statistical design in which we pooled the number of events observed in all animals, rather than a direct comparison of Puro+ versus Puro-, in a single animal. By this approach, we may underestimate statistical significance, but we obtain relationships that may have a general significance. First, we found a strong positive relationship between puromycin incorporation and Ki67 staining, suggesting clonal amplification (proliferation) of actively translating cells (*Figure 4C*, *Figure 4—source data 1*). Notably, we did not detect statistical differences in the expression of PD-1, TIGIT, CTLA-4, CD28, CD69, CCR5, CD103,

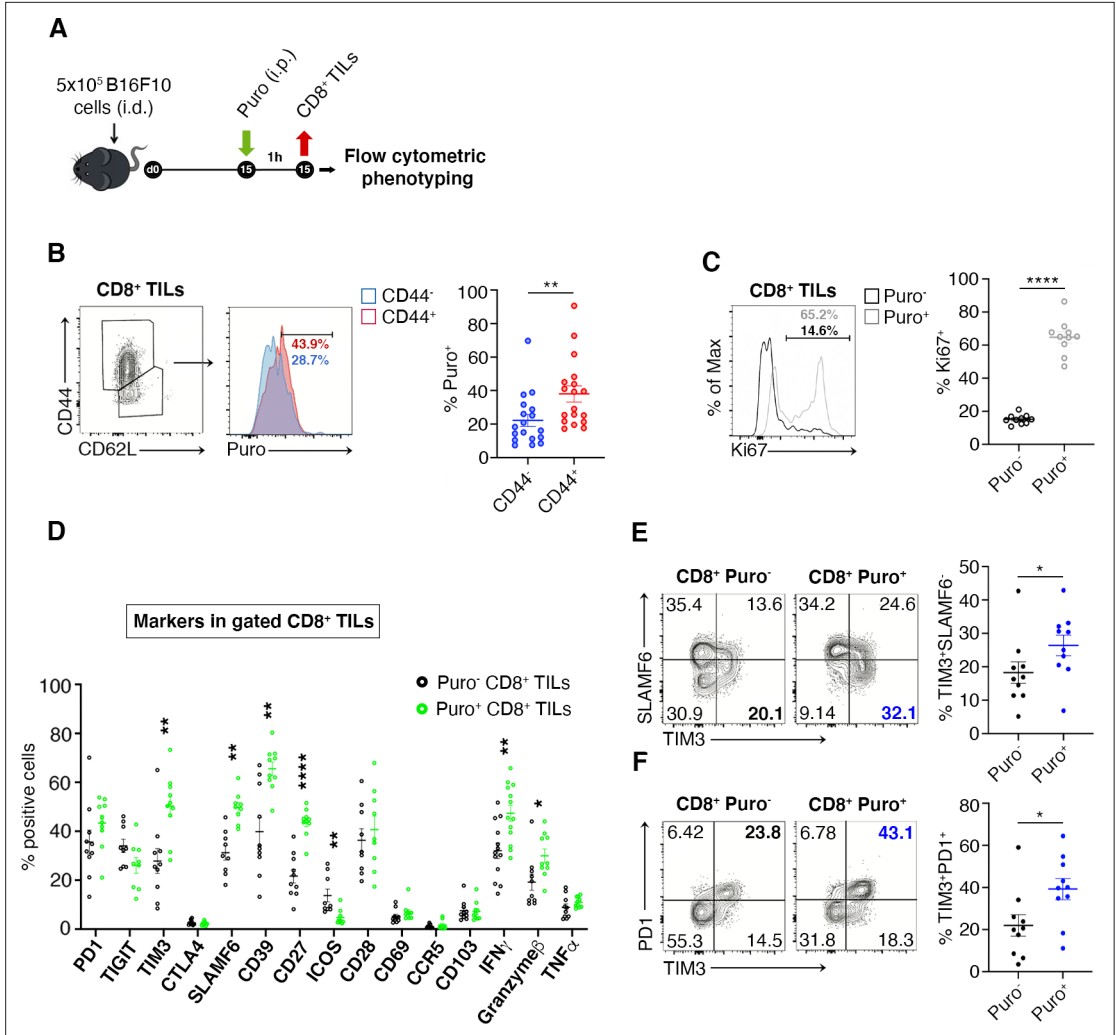

**Figure 4.** Puro+ translating CD8+ tumor-infiltrating lymphocytes (TILs) retain an activated-like phenotype. (**A**) Schematic diagram for the experiment. (**B**) Representative plots (left) and statistical analysis (mean ± SEM) of gated CD44+ CD8+ TILs analyzed for puromycin incorporation. Quantitation shows an enrichment for the expression of CD44. Percentages of positive cells in each gate are shown. Data from three experiments pooled together (n = 6 mice per experiment) **p < 0.01. (**C**) Representative plots (left) and statistical analysis (mean ± SEM) of Ki67+ within Puro+ and Puro- CD8+ TILs. Quantitation shows a positive correlation between translation rate and proliferation. Percentages of positive cells in each gate are shown. Data from two experiments pooled together (n = 5 mice per experiment) **p < 0.01. (**D**) Statistical analysis (mean ± SEM) for the indicated markers within Puro+ and Puro- CD8+ TILs shows that Puro+ translating CD8+ TILs are enriched for TIM3, SLAMF6, CD39, CD27, ICOS, and IFN-γ expression. Percentages of positive cells in each gate are shown. Data from two experiments pooled together (n = 5 mice per experiment) **p < 0.01; ****p < 0.0001. (**E**) Representative plots and statistical analysis (mean ± SEM) for SLAMF6 and TIM3 within Puro+ and Puro- CD8+ TILs shows that Puro+ translating CD8+ TILs are enriched for TIM3 expression. Percentages of positive cells in each gate are shown. Data from two experiments pooled together (n = 5 mice per experiment) *p < 0.05. (**F**) Representative plots and statistical analysis (mean ± SEM) for PD1 and TIM3 within Puro+ and Puro- CD8+ TILs show that Puro+ translating CD8+ TILs are enriched for TIM3 and PD1 expression. Percentages of positive cells in each gate are shown. Data from two experiments pooled together (n = 5 mice per experiment) *p < 0.05.

The online version of this article includes the following figure supplement(s) for figure 4:

**Source data 1.** Source data for *Figure 4B–F*.

**Figure supplement 1.** Flow cytometry analysis of PD1 and TIM3 within Ki67- puro+ and Ki67+ puro+ CD8+ tumor-infiltrating lymphocytes (TILs).

**Figure supplement 1—source data 1.** Source data for *Figure 4—figure supplement 1A, B*.

and TNF-α between high and low puromycin incorporating cells (*Figure 4D*, *Figure 4—source data 1*). Conversely, we found a positive relationship between puromycin incorporation and the expression of TIM3, SLAMF6, CD39, Granzyme B, CD27, and IFN-γ. In turn, ICOS expression was selectively decreased in highly translating CD8+ cells (*Figure 4D*). Next, we analyzed co-marker expression. We

found that highly translating cells were characterized by having high TIM3-low SLAMF6, high TIM3-high PD-1 (*Figure 4E–F*, *Figure 4—source data 1*). Consistently, high PD1 and high TIM3 expression associated with Ki67+ cells (*Figure 4—figure supplement 1A-B*, *Figure 4—figure supplement 1—source data 1*). Altogether, these data show that translationally active CD8+ T cells are confined in non-hypoxic areas, characterized by the activation of the mTORC1 pathway and can be defined by a specific immunophenotype. Taken together the data suggest the existence of clonally expanding CD8+ cells, highly translating, and moving versus the exhaustion status.

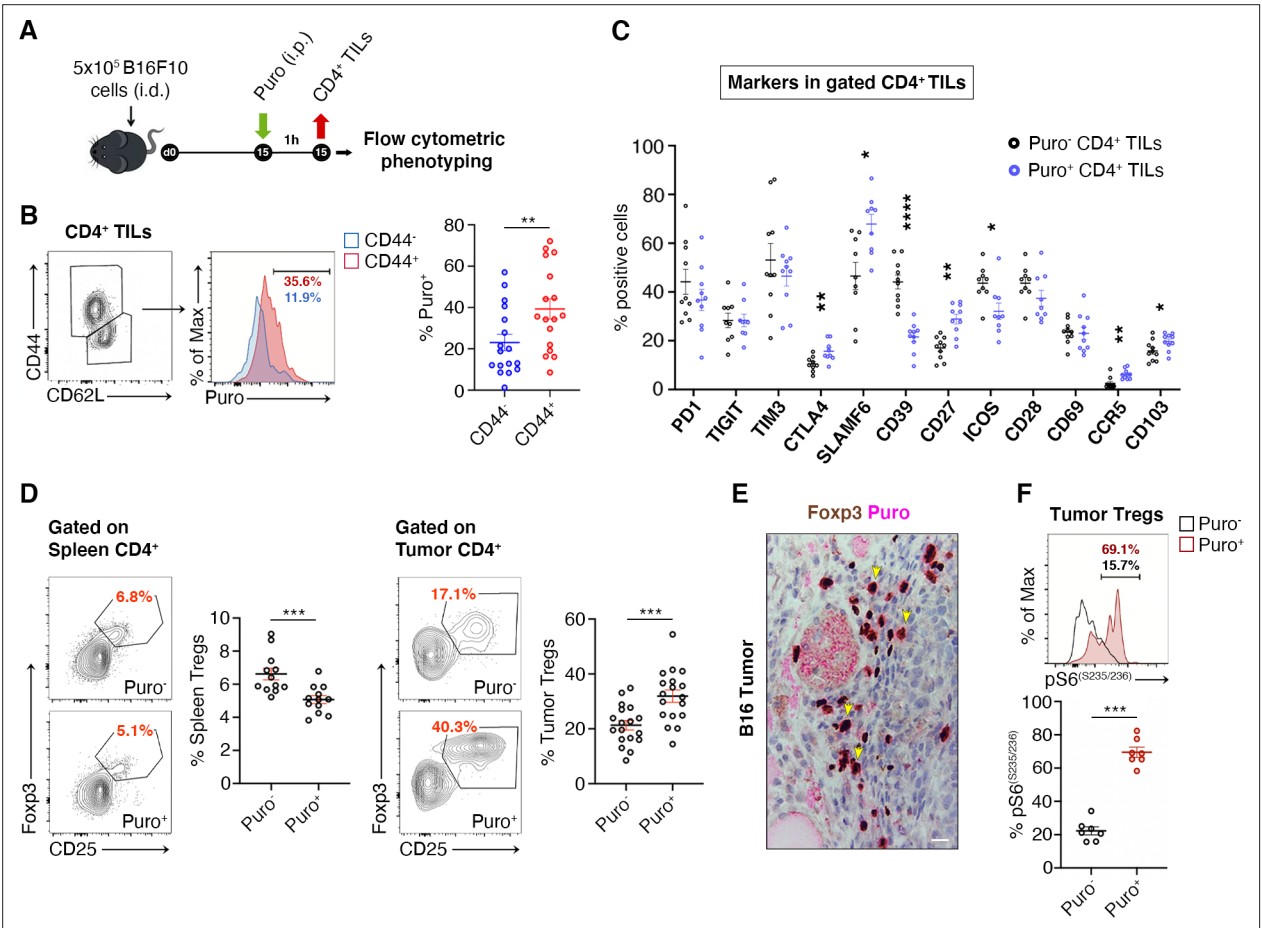

**Figure 5.** Regulatory T cells (Tregs) represents the majority of CD4+ tumor infiltrating Puro+ cells. (**A**) Schematic diagram for the experiment. (**B**) Representative plots (left) and statistical analysis (mean ± SEM) of gated CD44+ CD4+ tumor-infiltrating lymphocytes (TILs) analyzed for puromycin incorporation. Quantitation shows an enrichment for the expression of CD44. Percentages of positive cells in each gate are shown. Data from three experiments pooled together (n = 6 mice per experiment) **p < 0.01. (**C**) Statistical analysis (mean ± SEM) for the indicated markers within Puro+ and Puro- CD4+ TILs shows that Puro+ translating CD4+ TILs are enriched for CTLA4, CD39, CD27, ICOS, CCR5, and CD103 expression. Data from two experiments pooled together (n = 5 mice per experiment) *p < 0.05; **p < 0.01; ****p < 0.0001. (**D**) Representative plots (left) and statistical analysis (mean ± SEM) of gated CD25+ Foxp3+ CD4+ lymphocytes show that the number of Puro+ Tregs is higher in tumors versus spleen. Percentages of positive cells in each gate are shown. Data from two experiments pooled together (n = 4–5 mice per experiment) ***p < 0.001. (**E**) Immunohistochemical analysis of puromycin in Foxp3+ TILs shows that clusters of highly translating Foxp3+ cells are found. Scale bar, 20 μm. (**F**) Representative plots (left) and statistical analysis (mean ± SEM) for pS6(S235/236) within Puro+ and Puro- Tregs showing that, in vivo, most of translating Tregs have the mTORC1-S6K pathway active. Percentages of positive cells in each gate are shown. Data from one experiment representative of two (n = 7) ***p < 0.001.

The online version of this article includes the following source data and figure supplement(s) for figure 5:

**Source data 1.** Source data for *Figure 5B, C, D, and F*.

**Figure supplement 1—source data 1.** Source data *Figure 5—figure supplement 1*.

**Figure supplement 1.** Flow cytometric analysis of the indicated markers in CD4+ and Tregs in tumor.

## Efficiently translating CD4+ TILs are enriched for CTLA-4 expression and suppressor Tregs

We asked whether translating Puro+ CD4+ T cells had specific phenotypes. Next, we measured by flow cytometry the expression of markers of memory and effector T cell differentiation, activation, as well as of relevant costimulatory and coinhibitory molecules (*Figure 5A*). Herein, we summarize the main findings. We observed an increase in puromycin incorporation in CD44+ cells (*Figure 5B*, *Figure 5— source data 1*). Next, we analyzed the expression of costimulatory and coinhibitory proteins, that is, PD-1, TIGIT, TIM3, CTLA-4, SLAMF6, CD39, CD27, ICOS, CD28, CD69, CCR5, and CD103 (*Figure 5C*, *Figure 5—source data 1*). Some of these markers clustered with puromycin incorporation as shown in *Figure 5C*. Ki-67 labeling was robustly enriched in cells incorporating higher levels of puromycin (*Figure 5—figure supplement 1A*).

Tregs can be identified by CD25-FoxP3. We analyzed the difference in translation between tumor infiltrating Tregs and spleen-resident Tregs. Notably, only 4–6% Tregs of the spleen incorporated puromycin (*Figure 5D*, *Figure 5—source data 1*), whereas approximately 40 % of tumor-infiltrating Tregs incorporated high levels of puromycin (*Figure 5D*, *Figure 5—source data 1*). In keeping with these findings, IHC on paraffin-embedded tumor sections showed that puromycin signals overlap with FoxP3 staining (*Figure 5E*). Measurement of intracellular levels of phosphorylated S6 ribosomal protein (pS6) in gated Puro+ and Puro- Tregs showed, in line with previous observations, that pS6 labeling partitioned with high levels of puromycin, both for rpS6 S235/S236 (*Figure 5F*, *Figure 5— source data 1*) and for rpS6 Ser240/244 (*Figure 5—figure supplement 1*, *Figure 5—figure supplement 1—source data 1*).

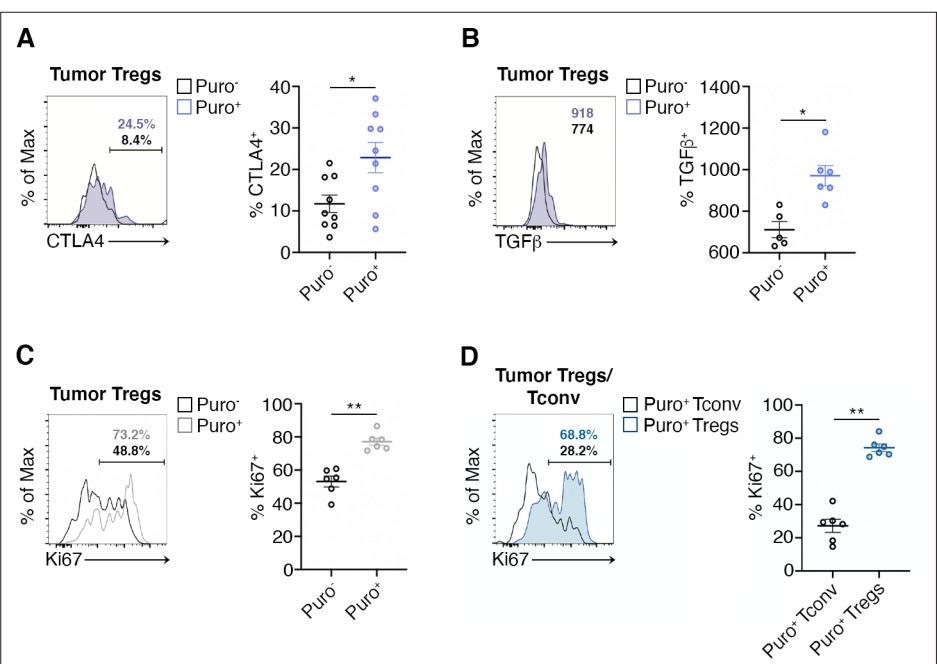

**Figure 6.** Highly translating regulatory T cells (Tregs) are functionally active and are more proliferative than highly translating Tconv. (**A**) Representative plots (left) and statistical analysis (mean ± SEM) for CTLA-4+ (**B**), TGF-β+ (**C**), Ki67+ (**C**), within Puro+ and Puro- Tregs shows that, in vivo, Puro+ translating Tregs are more proliferative but also exhibit a more activated phenotype than Puro- Tregs. Percentages of positive cells in each gate are shown. Data from one experiment representative of two (n = 6–9) *p < 0.05; **p < 0.01. (**D**) Representative plots (left) and statistical analysis (mean ± SEM) for Ki67+ within Puro+ Tregs and Puro+ Tconv shows that, in vivo, Puro+ Tregs expressed significantly more Ki67 than Puro+ Tconv (**C**), indicating that the former are more proliferative than the latter. Percentages of positive cells in each gate are shown. Data from one experiment representative of two (n = 6) **p < 0.01.

The online version of this article includes the following figure supplement(s) for figure 6:

**Source data 1.** Source data for *Figure 6A–D*.

CTLA-4 and TGF-β are Treg markers that correlate with the suppression activity of Tregs (*Sakaguchi et al., 2020*). We therefore gated Tregs and analyzed the correlation between puromycin incorporation, CTLA-4 and TGF-β. Puro⁺ Tregs showed the selective upregulation of CTLA-4 with respect to the Puro⁻ counterpart (*Figure 6A*, *Figure 6—source data 1*). Intracellular staining for TGF-β revealed enhanced secretion by Puro⁺ Tregs (*Figure 6—source data 1*, ). Puro⁺ Tregs had also higher levels of Ki67 staining (*Figure 6C*, *Figure 6—source data 1*). Overall, our data confirm the hypothesis that, within tumors, the main translating and expanding CD4⁺ subset is represented by suppressive Tregs with mTORC1 activation. If the hypothesis is true, then the expression of Ki67 should be higher in Puro⁺ Tregs, compared to Puro⁺ Tconv. Indeed, Puro⁺ Tregs expressed significantly more Ki67 than Puro⁺ Tconv cells (*Figure 6D*, *Figure 6—source data 1*), confirming that the tumor microenvironment preferentially fosters cell cycling of immunosuppressive cells.

## Discussion

TILs have been characterized by several transcriptomic studies leading to their extensive multi-lineage phenotypic classification (*Paijens et al., 2021*). In this study, we found that the translational activity of TILs is not uniformly distributed, being more intense in CD8⁺ T cells and CD4⁺ Tregs. A stratification of the translational capability of T cells associated with phenotypic markers shows that highly translating T cells are characterized by the presence of cytotoxic markers for CD8⁺ and suppressive markers for CD4⁺. Finally, highly translating T cells are characterized by markers for active cycling. The relevance of our observations is consistent with clinical evidence indicating that CD8⁺/Treg ratio is, in general, a strong predictive parameter for survival in cancer (*Gooden et al., 2011*). We will discuss some issues derived from our data.

Translation is strongly regulated by microenvironmental cues and its impairment acts as a limiting factor in tumorigenesis and tumor growth (*Barna et al., 2008*; *Miluzio et al., 2011*). Consequently, tumor genetic lesions converge on the translational machinery, leading to its constitutive activation, and rendering translation relatively independent of the signals posed by the microenvironment (*Loreni et al., 2014*; *Robichaud et al., 2019*). The relevance of the translational machinery in the cell autonomous growth of tumors is demonstrated by evidence indicating that genetic depletion of translation factors greatly reduces tumorigenesis and tumor growth (*Barna et al., 2008*; *Miluzio et al., 2011*; *Truitt et al., 2015*). In the tumor microenvironment, tumor cells are therefore taking full advantage of the restrictive nutrient conditions, thanks to the activation of oncogenic mutations. In sharp contrast, TILs lack mutations that alter their capability to synthesize proteins in conditions of high stress, and as such, are bound to permissive conditions in order to translate their transcriptional repertoire. In this context, one important question is whether the unfavorable conditions of the tumor microenvironment are affecting the action of some TILs classes, and whether they lead to specific developmental trajectories.

Hypoxia, far from being surprising, seems to act as a general inhibitor of translation of TILs. It is known since long that hypoxia inhibits translation acting on AMPK (*Horman et al., 2002*) and eIF2α (*Koumenis et al., 2002*). Phosphorylation of eIF2α occurs through four distinct kinases, PERK, GCN2, PKR, and HRI, that, together, are part of the integrated stress response, a complex set of events that drives, among others, the shut-off of general translation and the activation of specific mRNA translation (*Rios-Fuller et al., 2020*). Importantly, all four kinases are expressed in T cells at the mRNA level (*Critchley-Thorne et al., 2007*). PERK is considered a major regulator of translational control of hypoxia (*van den Beucken et al., 2006*). Unfortunately, we were not able to efficiently detect the specific activity of single eIF2α kinases in TILs, therefore it remains unresolved whether also in our conditions, PERK is the major regulator of translational shut-off and of eIF2α phosphorylation. It is possible that other pathways can contribute to the translational shut-off, given the sequel of adapting responses, especially in chronic conditions (*Schito and Semenza, 2016*), and considering that amino acid limitations affect TCR signaling (*Ron-Harel et al., 2019*). Oxygen sensing in the immune microenvironment shapes immunological responses, very often with confounding reports. In the past, it has been proposed that hypoxia leads to the preferential differentiation of peripheral, intratumor Tregs (*Dang et al., 2011*). In light of what we observe, it is unlikely that Treg in hypoxic environment can contribute to immune suppression, unless a sequel of phenomena occurs: first, the hypoxic environment causes conversion of conventional T cells in Tregs, which then persist for some time; second, the hypoxic environment induces angiogenesis, thus reviving the suppression and translation capability

of converted Tregs. It is obviously possible that a slow adaptation and phenotypic change can occur in human tumors, where the equilibrium between the immune system and tumor growth may last years. Summarizing, we think that our data are more consistent with a model in which the expansion of suppressive, intratumoral Tregs requires active translation and simultaneous increased rate of fatty acid synthesis (*Pacella et al., 2018*) and, in general, is associated with high glycolytic capability (*Procaccini et al., 2016*). In contrast, hypoxia may acutely impair the suppressive capability of Tregs.

The pathways that connect to translation are several, and their description is beyond the limits of this paper (*Roux and Topisirovic, 2018*). Our data indicate that the mTORC1 pathway is a major controller of TIL translation. In addition, we found that the Mnk/eIF4E pathway even if it was not massively detectable in most T cells, led to a subset of p-eIF4E positive cells correlating with high puromycin incorporation. From this perspective, it will be interesting to characterize subset of highly translating cells, in relationship to the activated pathways. This effort may require the establishment of new technologies to combine the detection of p-eIF4E with sequencing strategies, but it will be important in order to understand the impact of Mnk inhibition on cancer growth (*Bramham et al., 2016*). We were unable to effectively characterize the eIF6 pathway in TILs, due to the absence of antibodies detecting its phosphorylation (*Ceci et al., 2003*). Overall, each lymphocyte can differentially regulate its transcriptional repertoire depending on the specific pathways that are activated.

The observation of a different phenotype between T cells incorporating high levels of puromycin and T cells incorporating low levels has far-reaching implications. The correlation between translation and Ki-67 expression is logically linked to the fact that cell cycle progression requires de novo protein synthesis to sustain cell growth. Several studies at the single-cell level, combining mRNAseq and TCR sequencing, have unveiled the clonal expansion of subset of T cells. Interestingly, one major expanding subset is the CD4+ Tregs. TCR sequencing of breast cancer-associated Tregs revealed that Tregs have little TCR sharing with conventional T cells, confirming that Tregs in tumors are mainly generated through local expansion (*Plitas et al., 2016*). Local expansion of Tregs is supported by our data according to which translating T cells are Ki-67 positive. In general, we did not find differences between the analysis of the B16 mouse model and the MC38 one. We speculate that the local expansion of a translating phenotype is a general phenomenon and it occurs in areas that are not hypoxic.

The phenotypic identity of translating cells extends beyond the co-expression of Ki-67 and puromycin incorporation. CD8+ cells with prolonged exposure to antigens enter a state of exhaustion characterized by the elevated expression of inhibitory receptors (e.g., PD-1, CTLA-4, TIM-3, TIGIT, and LAG3). TILs expressing markers of exhaustion (PD-1, LAG3, and TIM3) are more likely to express IFN-γ (*Gros et al., 2014*). While exhausted CD8+ T cells have reduced function as compared to those elicited by an acute infection, several data suggest that are the exhausted CD8+ T cells that are exerting residual control over tumor growth (*Li et al., 2019*). Exhausted CD8+ T cells still play a critical role in cancer. Indeed, it is still confused how exhaustion is related to various properties, such as cell proliferation and effector functions. According to the operational definition given above, exhaustion is clearly not associated with the loss of translational activity, which results, for instance, in the enrichment of Ki-67 and TIM3 expression in Puro+ cells. The higher expression of CTLA-4 and TGF-β (*Sakaguchi et al., 2020*) in CD4+ translating cells fully correlates with suppressive properties.

In conclusion, the tumor microenvironment has been shown, in decades of studies, to provide a full range of stimuli (*Maman and Witz, 2018*) that may regulate translation, like hypoxia or nutrients. We suggest that two T cells that have an identical 'transcriptome' and protein repertoire infiltrate a tissue in two different microenvironment niches and are subject to rapid and differential translational regulation. If translational regulation persists, these two cells will have different developmental trajectories, as clearly shown by our study in which high puromycin incorporation goes hand in hand with the expression of specific markers. Last, our data suggest that T cell receptor stimulation and hypoxia are two main microenvironment clues that regulate the T cell trajectory. This further layer of complexity, however, rather than increasing the number of players in the process of tumor immunoediting enlightens the opposing role of CD8+ cytotoxic lymphocytes and CD4+ Tregs. Further characterization of translational control in these cells is required.

Limitations of our study: Quantitative puromycin incorporation was used to arbitrarily discriminate two T cell populations, highly translating and lowly translating. It is evident that this rough subdivision may not completely describe the tumor microenvironment. It is also evident that time may change the translational profile of a T cell, and a highly translating cell may shift to a lowly translating one if

the environmental conditions change: accordingly, we have currently no clues on whether the highly translating phenotype is stable.

# Materials and methods

**Key resources table**

| Reagent type (species) or resource | Designation | Source or reference | Identifiers | Additional information |
|---|---|---|---|---|
| Cell line (*Mus musculus*) | B16F10 | PMID:32699136 | | Cells were obtained from Dr Matteo Bellone, HSR Scientific Institute, Milan, Italy |
| Cell line (*Mus musculus*) | MC38 | PMID:32699136 | | Cells were obtained from Dr Maria Rescigno, Humanitas University, Rozzano (MI), Italy |
| Antibody | Mouse monoclonal anti-Puromycin, clone 12D10 | Millipore | Cat#: MABE343, RRID:AB_2566826 | WB (1:10000), IHC (1:500) |
| Antibody | Mouse monoclonal anti-Vinculin, clone V284 | Millipore | Cat#: 05–386, RRID:AB_11212640 | WB (1:1000) |
| Antibody | Mouse monoclonal anti-Actin | Sigma-Aldrich | Cat#: A4700, RRID:AB_476730 | WB (1:1000) |
| Antibody | Rabbit polyclonal anti phospho-rpS6 Ser235/236 | Cell Signaling | Cat#: 2211, RRID:AB_331679 | WB (1:1000) |
| Antibody | Rabbit polyclonal anti phospho-rpS6 Ser240/244 | Cell Signaling | Cell Signaling Technology Cat#: 2215, RRID:AB_331682 | WB (1:1000) |
| Antibody | Rabbit polyclonal anti-phospho-eIF2α Ser51 | Cell Signaling | Cat#: 3597, RRID:AB_390740 | IF (1:400), ELISA (1:500) |
| Antibody | Rabbit monoclonal anti-FoxP3 | Cell Signaling | Cat#: 98377, RRID:AB_2747370 | IHC (1:100) |
| Antibody | Rabbit monoclonal anti-CD4 | Cell Signaling | Cat#: 25229, RRID:AB_2798898 | IHC (1:100) |
| Antibody | Rat monoclonal anti-CD4 | BioLegend | BioLegend Cat#: 100401, RRID:AB_312686 | IF (1:400) |
| Antibody | Rabbit monoclonal anti-CD31 | Cell Signaling | Cat#: 77699, RRID:AB_2722705 | IHC (1:100) |
| Antibody | Mouse monoclonal APC conjugated anti-CD62L | eBioscience | Clone: MEL-14, Cat#: 17-0621-83 | FC (1:200) |
| Antibody | Mouse monoclonal PE conjugated anti-CCR5 | eBioscience | Clone: 7A4, Cat#: 12-1951-82 | FC (1:200) |
| Antibody | Mouse monoclonal FITC conjugated anti-ICOS | eBioscience | Clone: C398.4A, Cat#: 11-9949-82 | FC (1:200) |
| Antibody | Mouse monoclonal PE/Cy7 conjugated anti-CD39 | eBioscience | Clone: 24DMS1, Cat#: 25-0391-30 | FC (1:200) |
| Antibody | Mouse monoclonal PerCP/EF710 conjugated anti-CD3 | eBioscience | Clone: 17A2, Cat#: 46-0032-80 | FC (1:200) |
| Antibody | Mouse monoclonal PE conjugated anti-phospho-S6 | eBioscience | Clone: cupk43k, Cat#: 12-9007-41 | FC (1:100) |
| Antibody | Mouse monoclonal FITC conjugated anti-CD44 | BioLegend | Clone: IM7, Cat#: 103008 | FC (1:200) |
| Antibody | Mouse monoclonal PE/Cy7 conjugated anti-CD25 | BioLegend | Clone: PC61, Cat# 102016 | FC (1:200) |
| Antibody | Mouse monoclonal APC/Cy7 conjugated anti-CD4 | BioLegend | Clone: RM4-5, Cat#: 100526 | FC (1:200) |

*Continued on next page*

*Continued*

| Reagent type (species) or resource | Designation | Source or reference | Identifiers | Additional information |
|---|---|---|---|---|
| Antibody | Mouse monoclonal PE/Cy7 conjugated anti-CD4 | BioLegend | Clone GK1.5, Cat#: 100422 | FC (1:200) |
| Antibody | Mouse monoclonal PE/Cy7 conjugated anti-CD3 | BioLegend | Clone: 145–2 C11, Cat#: 100320 | FC (1:200) |
| Antibody | Mouse monoclonal Pacific Blue conjugated anti-CD8α | BioLegend | Clone: 53–6.7, Cat#: 100725 | FC (1:200) |
| Antibody | Mouse monoclonal PE conjugated anti-CTLA-4 | BioLegend | Clone: UC10-4B9, Cat#: 106305 | FC (1:100) |
| Antibody | Mouse monoclonal APC conjugated anti-Tim3 | BioLegend | Clone: B8.2C12, Cat#: 134007 | FC (1:200) |
| Antibody | Mouse monoclonal PerCP/Cy5.5 conjugated anti-PD1 | BioLegend | Clone: RMPI-30, Cat#: 109120 | FC (1:200) |
| Antibody | Mouse monoclonal PerCP/Cy5.5 anti-CD27 | BioLegend | Clone: LG.3A10, Cat#: 124214 | FC (1:200) |
| Antibody | Mouse monoclonal FITC conjugated anti-CD28 | BioLegend | Clone: E18, Cat#: 122007 | FC (1:200) |
| Antibody | Mouse monoclonal PE conjugated anti-CD103 | BioLegend | Clone: 2E7, Cat#: 121406 | FC (1:200) |
| Antibody | Mouse monoclonal PE/Cy7 conjugated anti-CD69 | BioLegend | Clone: H1.2F3, Cat#:104526 | FC (1:200) |
| Antibody | Mouse monoclonal PE/Cy7 conjugated anti-TIGIT | BioLegend | Clone: 1G9, Cat#:142107 | FC (1:200) |
| Antibody | Mouse monoclonal PE conjugated anti-SLAMF6 | BioLegend | Clone: 330-AJ, Cat#:134605 | FC (1:200) |
| Antibody | Mouse monoclonal APC conjugated anti- FoxP3 | eBioscience | Clone: FJK-16s, Cat#: 17-5773-82 | FC (1:100) |
| Antibody | Mouse monoclonal PE conjugated anti-Ki67 | eBioscience | Clone: SolA15, Cat#: 12-5698-80 | FC (1:100) |
| Antibody | Mouse monoclonal RedMab 549 conjugated anti-PMO | Hypoxiprobe | Cat#: 5914 | FC (1:100) |
| Antibody | Mouse monoclonal APC of FITC conjugated anti-Puromycin | Millipore | Clone: 12D10 | FC (1:100) |
| Antibody | Mouse monoclonal PE conjugated anti-IFN-γ | BioLegend | Clone: XMG1.2 Cat#: 505808 | FC (1:100) |
| Antibody | Mouse monoclonal APC conjugated anti- TNF-α | eBioscience | Clone: MP6-XT22, Cat#: 506308 | FC (1:100) |
| Antibody | Mouse monoclonal APC conjugated anti-Granzyme B | BioLegend | Clone: GB11, Cat#: 515303 | FC (1:100) |
| Antibody | Mouse monoclonal BV421 conjugated anti-TGF-β | Biolegend | Clone: TW7-16B4, Cat#: 141407 | FC (1:100) |
| Commercial assay or kit | BD Fixation Permeabilization Kit | eBioscience | Cat#: 554714 | |
| Commercial assay or kit | eBioscience FoxP3 staining buffer set | eBioscience | Cat#:: 00-5523-00 | |

*Continued on next page*

*Continued*

| Reagent type (species) or resource | Designation | Source or reference | Identifiers | Additional information |
|---|---|---|---|---|
| Commercial assay or kit | CD4$^+$ T Cell Isolation Kit | Miltenyi Biotec | 130-091-155 | |
| Chemical compound, drug | Human T-Activator CD3/CD28 | Thermo Fisher Scientific | 11131D | |
| Chemical compound, drug | Puromycin | Sigma-Aldrich | P8833 | |
| Chemical compound, drug | PMO | Hydroxyprobe | Cat#: HP1-XXX | |
| Chemical compound, drug | Everolimus | Sigma-Aldrich | SML2282 | |
| Chemical compound, drug | PPP242 | Sigma-Aldrich | P0037 | |
| Chemical compound, drug | MNK inhibitor | Sigma-Aldrich | 454861 | |

## Human samples and mice

Human blood samples from healthy male or female donors were collected with written informed consent, and collection was performed according to protocols approved by Ethics Committee of Fondazione Istituto di Ricovero e Cura a Carattere Scientifico Ca'Granda Ospedale Maggiore Policlinico. All animal experiments were performed in accordance with the Swiss Federal Veterinary Office guidelines and approved by the Ethical Committee of the Cantonal Veterinary with authorization number TI 37/2016. C57BL/6 J mice were bred in specific pathogen-free facility at the Institute for Research in Biomedicine (Bellinzona, Switzerland). Mice were housed, five per cage, in ventilated cages under standardized conditions (20 °C ± 2 ° C, 55% ± 8% relative humidity, and 12 hr light/dark cycle). Food and water were available ad libitum, and mice were examined daily.

## Cell isolation and culture from human blood

PBMCs were isolated from blood samples by Ficoll-Hypaque density-gradient centrifugation. CD4$^+$ T cells were enriched from PBMCs by magnetic separation (AutoMACS, Miltenyi Biotec) using human CD4$^+$ T Cell Isolation Kit (Miltenyi Biotec) according to the manufacturer's instructions before flow sorting on a FACSAria II (BD Biosciences).

CD4$^+$ T cells were activated with anti-CD3/CD28-coated beads (Life Technologies Dynabeads T-Activator) and IL-2 (20 IU/mL) and cultured in RPMI 1640 medium (Life Technologies) supplemented with 1 % penicillin-streptomycin (Life Technologies), 2 mM GlutaMAX (Life Technologies), 1 mM sodium pyruvate (STEMCELL Technologies) for the indicated time intervals.

Typical cell culture conditions (37 °C, 5 % $CO_2$, and environmental [21%] oxygen) were used and the conditions referred to as normoxia. Hypoxia was generated with an oxygen/$CO_2$ controller incubator (Galaxy 48 R; Eppendorf). It was set at 1 % oxygen and 5 % $CO_2$.

## Cell isolation from mice specimens

T cells were obtained from cell suspensions of spleen. For TILs isolation, tumors were cut in small pieces and resuspended in RPMI1640 with 1.5 mg/mL Type I Collagenase (Sigma), 100 mg/mL DNase I (Roche), and 5 % FBS, digested for 45 min at 37 °C under gentle agitation. The digestion product was then passed through a 70 µm cell strainer to obtain a single-cell suspension. Lymphocytes were then enriched by Percoll density gradient following the manufacturer's protocol.

## Murine tumor cell lines

B16F10 and MC38 cells were cultured in RPMI1640 supplemented with 10 % heat-inactivate FBS, 100 U/mL penicillin/streptomycin, and 100 U/mL kanamycin. Cells were tested for the absence of *Mycoplasma* and maintained in 5 % $CO_2$ at 37 °C. Frozen aliquots were thawed for each in vivo

experiment and passaged in vitro for the minimum time required. Tumor cells at 70–80% confluency were harvested by diluting them 1:5 in 0.25 % trypsin.

## In vitro measurement of protein synthesis

For in vitro measurements of protein synthesis, CD4$^+$ T cells stimulated in vitro for the indicated time intervals were treated with 5 µg/mL puromycin for 10 min. Puromycin incorporation was then determined by flow cytometry or Western blotting as described later.

## In vivo measurement of protein synthesis

To measure protein synthesis in vivo, mice were intraperitoneal injected with puromycin at a concentration of 50 mg/kg dissolved in PBS at pH 6.4–6.6; 1 hr later, mice were sacrificed, and mice specimens rapidly collected. Spleen or cutaneous tumors were further processed for flow cytometry or histology analysis as described later. The sample size was estimated according to the following parameters: power 80%, alpha 0.05, average means of Puro$^+$ cells 60% ± 20% for the tumor group and 30 % for the non-tumor sample. Consequently, the minimal number was fixed to N = 5/group.

For the analysis of the effect of everolimus on translation rate and mTORC1 signaling, B16 mice were given intraperitoneal injections of 5 mg/kg everolimus (Sigma-Aldrich) dissolved in PBS for 2 consecutive days. Puromycin was injected 1 hr before sacrificing mice and collecting TILs for subsequent flow cytometry analysis.

## In vitro stimulation of lymphocytes and hypoxia

CD4$^+$ T cells stimulated in vitro for 36 hr were transferred from 20 % $O_2$ to 1 % $O_2$ for the indicated times. Translation and eIF2α phosphorylation were measured by flow cytometry and ELISA, respectively, as described later.

## In vivo detection of tissue hypoxia

PMO (Hydroxyprobe) was injected intraperitoneally at a concentration of 60 mg/kg dissolved in PBS at pH 6.4–6.6. After 1 hr, mice were killed and processed further for flow cytometry or fluorescence microscopy analysis, as described later.

## In vitro eIF2α phosphorylation detection with ELISA assay

Phosphorylation of eIF2α was measured by ELISA assay. For the assay, each well of a 96-well microtiter plate was coated overnight at 4 °C in a humidified chamber with 20 µg of protein extracts of stimulated CD4$^+$ T cellsì either incubated for 4 hr under normoxia or hypoxia. Plate was then probed with either rabbit anti-total-eIF2α antibody (1:500, Cell Signaling) or anti-phospho-specific eIF2α$^{(S51)}$ antibody (1:500, Cell Signaling), for 1 hr at room temperature, followed by incubation with HRP-conjugate secondary antibody for 30 min at room temperature and addition of ELISA colorimetric TMB reagent as a soluble substrate for the detection of peroxidase activity. The absorption intensity was obtained using the Model 680 Microplate Reader (Bio-Rad).

## Histology, tissue stainings, antibodies, and imaging

Spleen or tumor samples were either embedded in OCT and frozen or fixed overnight with 4 % paraformaldehyde, transferred to 70 % EtOH and embedded in paraffin as previously described (*Rosso et al., 2004*). Samples were then cut at 4 µm (paraffin) or 10 µm (frozen) and stained with hematoxylin and eosin (Sigma-Aldrich) for morphological analysis. Immunohistochemical and immunofluorescence analyses were performed as described previously (*Manfrini et al., 2020a*).

Primary antibodies were used at the following dilutions: mouse monoclonal to Puromycin 12D10 (1:500, Millipore), rabbit polyclonal to phospho-eIF2α$^{(S51)}$ (1:400, Cell Signaling), rabbit monoclonal to FoxP3 D2W8ETM (1:100, Cell Signaling), rabbit monoclonal to CD31 D8V9E (1:100, Cell Signaling), rabbit monoclonal to CD4 D7D2Z (1:100, Cell Signaling), rat monoclonal to CD4 GK1.5 (1:400, BioLegend). Secondary Alexa Fluor conjugated goat anti-mouse, donkey anti-rat, and donkey anti-rabbit antibodies were used at the 1:500 dilution (Thermo Fisher Scientific).

Slides were mounted in glycerol supplemented with Mowiol 4–88 mounting medium (Sigma-Aldrich). White field images were acquired using a Leica DM1600 microscope. Fluorescence images

were acquired using a confocal microscope (Leica TCS SP5) at 1024 Å, ~1,024 dpi resolution. All the images were further processed with Photoshop CS6 (Adobe, Berkeley, CA) software.

## Western blotting

SDS-PAGE was performed on protein extracts obtained from human CD4$^+$ T cell activated in vitro as previously described. Western blotting was performed as previously described (*Ricciardi et al., 2018*). The following antibodies were used for Western blotting: rabbit polyclonal to phospho-rpS6 Ser235/236 (1:1000, Cell Signaling), rabbit polyclonal to phospho-rpS6 Ser240/244 (1:1000, Cell Signaling), mouse monoclonal anti-Vinculin (1:1000, Millipore), mouse monoclonal anti-Actin (1:1000, Sigma), mouse monoclonal anti-Puromycin (1:10000, Millipore). Chemiluminescent signals were detected using Amersham ECL Prime (GE Healthcare Life Sciences) and images were acquired using the iBright CL750 Imaging System (Thermo Fisher Scientific).

Where indicated, cells were treated with either 2 µM PP242 (Sigma-Aldrich) or 3 µM MNK inhibitor (Sigma-Aldrich) for 30 min after 48 hr of Dynabeads stimulation.

## Flow cytometry

For assessment of puromycin incorporation on human blood-derived samples, CD4$^+$ T cells stimulated in vitro for the indicated times were fixed and permeabilized with Cytofix and Perm Buffer III buffers (both BD Biosciences) according to the manufacturer's protocol, and stained with FITC conjugated anti-Puromycin antibody (Millipore). Finally, cells were analyzed by flow cytometry, and the relative amount of protein synthesized quantified with FlowJo software.

The following anti-mouse mAbs were purchased from eBioscience: APC conjugated anti-CD62L (clone: MEL-14, Cat#: 17-0621-83), PE conjugated anti-CCR5 (clone: 7A4, Cat#: 12-1951-82), FITC conjugated anti-ICOS (clone: C398.4A, Cat#: 11-9949-82), PE/Cy7 conjugated anti-CD39 (clone: 24DMS1, Cat#: 25-0391-30) PerCP/EF710 conjugated anti-CD3 (clone: 17A2, Cat#: 46-0032-80), PE conjugated anti-phospho-S6 (clone: cupk43k, Cat#: 12-9007-41).

The following mAbs were purchased from BioLegend: FITC conjugated anti-CD44 (clone: IM7, Cat#: 103008), PE/Cy7 conjugated anti-CD25 (clone: PC61, Cat# 102016), APC/Cy7 conjugated anti-CD4 (clone: RM4-5, Cat#: 100526), PE/Cy7 conjugated anti-CD4 (clone GK1.5, Cat#: 100422) PE/Cy7 conjugated anti-CD3 (clone: 145–2 C11, Cat#: 100320), Pacific Blue conjugated anti-CD8α (clone: 53–6.7, Cat#: 100725) PE conjugated anti-CTLA-4 (clone: UC10-4B9, Cat#: 106305), APC conjugated anti-Tim3 (clone: B8.2C12, Cat#: 134007), PerCP/Cy5.5 conjugated anti-PD1 (clone: RMPI-30, Cat#: 109120), PerCP/Cy5.5 anti-CD27 (clone: LG.3A10, Cat#: 124214), FITC conjugated anti-CD28 (clone: E18, Cat#: 122007), PE conjugated anti-CD103 (clone: 2E7, Cat#: 121406), PE/Cy7 conjugated anti-CD69 (clone:H1.2F3, Cat#: 104526), PE/Cy7 conjugated anti-TIGIT (clone: 1G9, Cat#:142107), PE conjugated anti-SLAMF (clone: 330-AJ, Cat#: 134605).

Intracellular staining was performed using the BD Cytofix/Cytoperm and Perm/Wash buffers or, for intracellular FoxP3 (APC, clone: FJK-16s, eBioscience, Cat#: 17-5773-82), Ki-67 (PE, clone: SolA15, eBioscience, Cat#: 12-5698-80), PMO (Hypoxiprobe Red Mab 549, Cat#5914), and Puromycin (APC or FITC, clone#12D10, Millipore) staining, the eBioscience FoxP3 staining buffer set. For intracellular staining of IFN-γ (PE, clone: XMG1.2, Biolegend, Cat#: 505808), TNF-α (APC, clone: MP6-XT22, eBioscience, Cat#: 506308), Granzyme B (APC, clone: GB11, Biolegend, Cat#: 515303), and TGF-β (BV421-labeled, clone:TW7-16B4, Biolegend, Cat#:141407) T cells were incubated for 4 hr at 37 °C in ionomycin (Sigma-Aldrich,750 ng/mL) and PMA (Sigma-Aldrich, 20 ng/mL). For the last 3 hr, Brefeldin (eBioscience, 1000 × Solution) was added to the cultures.

## Statistical analysis

The data are expressed as mean ± SEM. Student't t test was used to compare pairs of data. In mice experiments, the statistical analysis was performed with the Prism Software (GraphPad). Comparisons of two groups were calculated using nonparametric Mann-Whitney test or ANOVA with Dunnett's post hoc test. Error bars, p values, and statistical tests are reported in the corresponding figure legends. All experiments were based on biological replicates, that is, different groups of mice or T cells from different donors. No samples were excluded from the analysis.

## Source data

Excel files are provided for FACS and ELISA analysis. Raw gels as *Figure 1—source data 1*. Gels with cropped images as *Figure 1—source data 2*.

## Acknowledgements

The generous contribution of Fondazione Invernizzi for the Centro Organoidi and of AIRC IG 19973 to SB is acknowledged. We thank for initial help with the immunohistochemistry Giulia Mensa. This manuscript is dedicated to the memory of Prof. Fabrizio Loreni.

## Additional information

### Funding

| Funder | Grant reference number | Author |
|---|---|---|
| Associazione Italiana per la Ricerca sul Cancro | IG 19973 | Stefano Biffo |
| Fondazione Romeo ed Enrica Invernizzi | 001 | Stefano Biffo |

The funders had no role in study design, data collection and interpretation, or the decision to submit the work for publication.

### Author contributions

Benedetta De Ponte Conti, Conceptualization, Data curation, Formal analysis, Investigation; Annarita Miluzio, Data curation, Investigation, Methodology; Fabio Grassi, Investigation, Supervision, Writing - original draft, Writing - review and editing; Sergio Abrignani, Conceptualization, Writing - review and editing; Stefano Biffo, Conceptualization, Project administration, Resources, Supervision, Writing - original draft, Writing - review and editing; Sara Ricciardi, Conceptualization, Data curation, Formal analysis, Supervision, Validation, Visualization, Writing - original draft, Writing - review and editing

### Author ORCIDs

Stefano Biffo (iD) http://orcid.org/0000-0002-3780-1050
Sara Ricciardi (iD) http://orcid.org/0000-0001-8124-432X

### Ethics

Human subjects: Informed consent, and consent to publish, was obtained. "Studio comparativo del sistema immunitario tissutale in patologietumorali e autoimmuni con tecnologie multi-omiche (immunom/2020)." September 15, 2020.

This study was performed in strict accordance with the recommendations of the Ethical Commette of the Cantonal Veterinary of Switzerland. All of the animals were handled according to approved institutional animal care and use committee (IACUC) protocols (TI 37/2016) of the Cantonal Veterinary of Switzerland. No surgery was performed. Animals were sacrificed by $CO_2$ euthanasia and every effort was made to minimize suffering.

### Decision letter and Author response

Decision letter https://doi.org/10.7554/eLife.69015.sa1
Author response https://doi.org/10.7554/eLife.69015.sa2

## Additional files

### Supplementary files

• Transparent reporting form

### Data availability

All data generated or analysed during this study are included in the manuscript and supporting files.

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
