## [Editor Report]

By employing melanoma and colon cancer allograft mouse models, the authors show that tumor-infiltrating T-cells exhibit heterogeneity in levels of protein synthesis that correlates with their immunophenotypes. Moreover, some evidence is provided that the observed heterogeneity in protein synthesis rates in tumor-infiltrating T-cells levels is driven by distinct conditions in different parts of tumor microenvironment. Overall, these findings further corroborate the importance of mRNA translation in immune cell plasticity and suggest that relying on monitoring steady-state mRNA levels may not provide the full picture of immunophenotypes in tumor niche.

---

## [Decision Letter]

**Decision letter after peer review:**

Thank you for submitting your article "mTOR-dependent translation drives tumor infiltrating CD8^+^ Effector and CD4^+^ Treg cells expansion" for consideration by *eLife*. Your article has been reviewed by 3 peer reviewers, including Ivan Topisirovic as Reviewing Editor and Reviewer #1, and the evaluation has been overseen by Carla Rothlin as the Senior Editor. The following individual involved in review of your submission has agreed to reveal their identity: Ola Larsson (Reviewer #2).

Essential revisions:

1) It was thought that the focus on mTOR signaling is too narrow and that other relevant pathways impinging on translation apparatus should be considered. These include integrated stress response and MNK/eIF4E axis, as these pathways have been implicated in adaptation to hypoxia and/or immune responses in neoplasia.

2) In some parts of the manuscript methodology/analysis should be improved. Specifically, anti-phospho-S240/244 ribosomal protein S6 antibody should be used instead of the antibody recognizing phosphorylated S235/236, as the latter phosphoacceptor sites are not exclusively phosphorylated in a mTOR-dependent manner. In addition, it was found that the statistical tests used were in some cases inappropriate, as illustrated in reviews below.

3) Assessing co-expression of the markers presented in figures 4D and 5D and relating their levels to protein synthesis activity was thought to be warranted in order to grasp full heterogeneity of tumor-infiltrating T cells.

4) Finally, it was thought that the manuscript may benefit from additional editing.

*Reviewer #1:*

In this study, mouse models are employed to systematically establish protein synthesis activity of tumor-infiltrating CD4^+^ and CD8^+^ T lymphocytes in vivo. The effects of stimuli emanating from tumor niche on translational activity of CD4^+^ and CD8^+^ T lymphocytes were also studied. Using ex vivo experiments, the evidence is provided that T cell receptor stimulation induces protein synthesis and mTOR signaling in tumor-infiltrating T cells, whereas hypoxia exerts the opposite effect. Next, the authors used mouse melanoma and colorectal cancer models to confirm these findings. Using the same models, the authors show that tumor-infiltrating CD4^+^ and CD8^+^ T lymphocytes translate more than their counterparts in the spleen. Importantly, it was also observed that tumor-infiltrating lymphocytes also exhibit significant intercellular heterogeneity in translational activity. This was followed by establishing the correlations between levels of protein synthesis and T cell phenotypes using flow cytometry, which revealed strong correlation between levels of protein synthesis and immunophenotypes. Collectively, these findings support the tenet that translation plays a major role in establishing tumor-infiltrating T cell phenotypes, and demonstrate that there is pronounced intracellular heterogeneity in their protein synthesis activity. This heterogeneity in the levels of mRNA translation between tumor-infiltrating T cells may at least in part be explained by the difference in stimuli that these cells are exposed in distinct parts of tumor microenvironment. In long-term, this study may inspire single-cell proteomics and/or similar approaches that do not rely on steady-state mRNA levels to monitor the plasticity of tumor-infiltrating lymphocytes and thus provide molecular bases for more efficient immunotherapies.

Strengths: Presented findings further support the critical role of the protein synthesis apparatus in establishing lymphocyte phenotypes, and emphasize the importance of considering alterations that occur downstream of changes in mRNA levels when determining the lymphocyte subtypes in the tumors. This is of particular interest considering recent increase in the number of single cell sequencing studies aiming to address composition of tumor-infiltrating lymphocytes, wherein correlations with phenotypes are driven based on steady-state mRNA composition of the cell. Methodologically, the clear advantage of this study, is that the most of the work was done in vivo, and is therefore of high physiological relevance. In addition, both melanoma and colon cancer models were employed, thereby suggesting that reported observations are likely to be generally applicable, and not cancer type-specific. In addition, the authors monitored the effects of cancer-relevant perturbations on protein synthesis, including T cell receptor stimulation and hypoxia. Overall, this study provides solid evidence that translation plays a major role in establishment of immunophenotypes of the tumor-infiltrating lymphocytes, and that interaction between tumor microenvironment and translational machinery may be responsible for intercellular heterogeneity of T cells in tumor niche.

Weaknesses: The major drawback of this study is that it relies on correlation and lacks experiments aiming to decipher mechanisms underlying the interactions between tumor niche and translational machinery of tumor infiltrating lymphocytes. Moreover, it was thought that the focus on mTOR may be too narrow and that other signaling mechanisms known to modulate translation rates in response to environmental cues should also be investigated. Finally, it remains unclear whether the changes in protein synthesis rates affect all the mRNAs uniformly or induce selective perturbations in translational activity of specific subsets of mRNAs.

Notwithstanding these weaknesses, it was thought that with appropriate revision, provided evidence is sufficient to support the major conclusion of the manuscript which is that the interactions between tumor microenvironment and translational machinery play a major role in determining immunophenotypes of tumor-infiltrating lymphocytes and that relying on steady-state mRNA levels may not be sufficient to grasp full plasticity of tumor-infiltrating T cells.

– Sole focus of the study is on mTOR signaling. This is somewhat surprising as other mechanisms have been also shown to play a central role in translational reprogramming under hypoxia and T cell receptor stimulation (e.g. ISR, MNK-dependent phosphorylation of eIF4E, etc). Based on this, it was thought that more comprehensive assessment of alterations in signaling pathways that impinge on translational machinery is warranted to pinpoint signaling pathways that mediate the observed effects of tumor microenvironment on T lymphocytes.

– Conclusions on alterations in mTOR signaling are based solely on monitoring phospho-rpS6 levels. Importantly, the antibody that the authors used recognizes S235/236 on rpS6 which are phosphorylated in cells lacking S6K1 and 2 (PMID: 15060135) and are phosphorylated by other AGC kinases (e.g. RSKs). The phoposho-acceptor site on rpS6 that is exclusively sensitive to alterations in the mTORC1/S6K signaling axis is S240/244 (PMID: 15060135) and thus, antibody recognizing S240/244 should be employed. To strengthen their conclusions, the authors should also add appropriate total antibody controls and monitor other mTORC1 targets (e.g. 4E-BPs, eEF2K).

– Considering the general lack of mechanistic data to explain heterogeneity in translational activity of tumor-infiltrating T cells, it was thought that monitoring the impact of mTOR inhibitors as well as compounds targeting other signaling mechanisms that adjust protein synthesis in accordance to changes in cellular environment (e.g. ISRIB, MNK inhibitors) on T-cell immunophenotypes and heterogeneity in vivo would be informative.

– It is well-established that in addition to changes in global protein synthesis levels, stressors such as hypoxia selectively affect translation of different mRNA subsets, whereby the latter changes are likely to significantly contribute to observed immunophenotypes. I understand that translatome-wide translational profiling is out of the scope of this short report and also technically very challenging, but perhaps authors should monitor mRNA and protein levels of selected number of factors that are known to be translationally regulated under these conditions and/or implicated in T cell biology in their system.

– The article may benefit from some careful editing.

*Reviewer #2:*

The tumor microenvironment includes the extracellular matrix (ECM) and a multitude of cells including immune cells. During cancer progression, the tumor associated ECM and cell composition/activity is altered. For example, the activity of immune cells towards tumor cells is commonly reduced which facilitates e.g. tumor growth and metastasis. It is well established that, in vitro, activation of T lymphocytes leads to a dramatic upregulation of protein synthesis and increased proliferation. Yet, to what extent this is true in vivo is not yet characterized. In this manuscript, Benedetta De Ponte Conti et. al address this using a FACS-based approach whereby global protein synthesis (quantified by puromycin incorporation) is related to expression of established activation markers for CD4^+^ and CD8^+^ T cells in tumors from the B16/F10 melanoma model. The conclusions are largely in agreement with studies of in vitro models as activation of CD4 or CD8 cells induces proliferate and bolsters protein synthesis. This study thereby pinpoints the close relationship between protein synthesis and immune cell activity also in vivo. One particularly interesting finding is the links between protein synthesis and Treg phenotype. In aggregate, the claims are well supported and important. This study will likely fuel many additional studies trying to in greater detail understand how protein synthesis in immune cells relate to their function within the tumor microenvironment.

There are some parts I think could be more developed:

1. There is a close link between protein synthesis and proliferation following activation of CD4 and CD8 T cells. This raises the question of how close puromycin incorporation associates with proliferation. In figure 5 S1, Ki67 vs protein synthesis is assessed in CD4^+^ T cells. This suggests a very close relationship. IS there a corresponding relationship for CD8^+^ T cells? Are there any aspects of the protein synthesis profile which do not associate with proliferation? What would the phenotype of such cells be?

2. All analyses are performed on populations of cells but there are no cross-comparisons between marker expression. I.e. is there one subset of T cells with high protein synthesis which express most of these markers (Figure 4D) or are there subsets of high protein synthesis cells which express subsets of markers? I am not sure if it is technical possible, but it would be interesting to assess co-expression of the identified markers in figure 4D and 5D and how their expression relates to protein synthesis.

3. Analysis is largely qualitative when assessing expression of makers and protein synthesis. Perhaps it would be informative to also perform a quantitative analysis within single tumors?

4. It is unclear to me what the relationship between T cell exhaustion and protein synthesis (and also proliferation) is based on the presented data. It seems logical to expect reduced protein synthesis in exhausted T cells, but this may not be the case.

*Reviewer #3:*

Overall the figures and the text were clear and concise and the data well described. The data shown support the claims made, but the authors have not examined the other major pathway for regulation of translation. My main issues relate to the first three figures.

1. The authors focus on the regulation of protein synthesis through mTOR, however translation initiation is also regulated by eIF2. Moreover, the PERK-eIF2-ATF4 axis has been shown to be important for tumour cell adaptation to stress – therefore, as all four eIF2 kinases are expressed in these models, it is important to determine any effect from eIF2 on translation activation/inhibition:

– What are the levels of eIF2α phosphorylation in the -/+puromycin cell populations (Figure 1 and 2)?

– As hypoxia is known to regulate eIF2 kinases what is the activation status of the four kinases in these model systems during hypoxia (Figure 1 – 3)?

– It is known that mTOR signalling can crosstalk to eIF2 signalling, and vice-versa, so how do the levels of eIF2α phosphorylation (and activity of eIF2 kinases) correlate with mTOR activity in these models during hypoxia (Figure 1 – 3)?

– To support the author's conclusions it would be necessary to show that the regulation of translation in this model is independent of eIF2 signalling.

2. The author's state at various points that the data shows mTOR activity is increased in the puromycin positive proportion of cells, however they do not directly show this:

– The antibody used for phospho-RPS6 recognises residues (Ser235/236), which can be phosphorylated in the absence of S6K1 and S6K2, therefore, could this regulation be independent of mTOR? Although RPS6 is downstream of mTOR signalling it is not a direct substrate of mTOR, therefore, to fully support these conclusions, blots for direct substrates of mTOR (p-S6K1, p-4EBP1), with corresponding blot for all total proteins, should be shown.

3. Figure 1C – Puromycin incorporation demonstrates that protein synthesis is increased yet only the phospho-RPS6 blot is included. A total RPS6 blot must be included to demonstrate that the increase in phospho-RPS6 signal is not simply increased synthesis of total RPS6. This is especially important as the abundance of vinculin (used as the loading control) varies across samples in Figure 1B and 1C.

4. Figure 1F – statistical analysis – multiple comparisons have been made to the 0 hr time point. Running multiple t-tests in this manner increases the chance of error, therefore, statistical analysis should take these multiple comparisons into account by using ANOVA with Dunnett's post hoc test.

---

## [Author Response]

Essential revisions:1) It was thought that the focus on mTOR signaling is too narrow and that other relevant pathways impinging on translation apparatus should be considered. These include integrated stress response and MNK/eIF4E axis, as these pathways have been implicated in adaptation to hypoxia and/or immune responses in neoplasia.

We understand. The focus on the mTORC1 pathway was due to the overwhelming role that it has within both immune system biology and translation, perhaps also for the existence of excellent reagents and a good comprehension of its basic function. We included statements on translation pathways in the Introduction and in the Discussion, and added several experiments.

To comply (experimentally) with this criticism, we have started to analyse in vitro in primary T cells the effects of: 1. hypoxia on Ser51 eIF2α phosphorylation, 2. the effects of Mnk inhibition on puromycin incorporation and on Ser209 eIF4E phosphorylation.

1. eIF2a phosphorylation was below reliable detection limit by Western Blotting in primary human T cells. Next, we attempted to set up an ELISA assay: the assay was successful for phospho-eIF2α and demonstrates an increase of eIF2α phosphorylation in hypoxia conditions (Figure 1G). Next, we attempted FACS analysis of eIF2α phosphorylation in mouse TILs. In our hands, we cannot detect in lymphocytes reliable, specific 2α phosphorylation by FACS analysis (Author response image 1). We then turned to immunohistochemistry: starting with the observation that PMO hypoxic labelling correlates with low puromycin incorporation (Figure 3D), we were able to set up triple staining methods for CD4^+^, PMO, P-eIF2α. We were able to demonstrate that PMO staining clusters with eIF2α staining in CD4^+^ cells (Figure 3 and Figure 3 —figure supplement 1). Overall, a negative correlation between puromycin and PMO staining/eIF2α/phosphorylation seems evident.

**Author response image 1. sa2fig1:** Representative flow cytometry data of unstained CD4+ TILs used as a negative control and labelled CD4+ TILs with Alexa Fluor 405 anti-human phospho-eIF2α proving the almost totally absence of a specific 2α phosphorylation signal.

2. eIF4E phosphorylation was below reliable detection limit by Western Blotting in primary human T cells. We also performed ELISA assay for phosho-eIF4E, but we did not see changes in eIF4E phosphorylation that we could confirm as specific. There are two explanations, (a) eIF4E is not phosphorylated in activated T cells, (b) the extent of phospo/dephospho eIF4E is below reliable detection, either because a minority of cells is affected or due to low levels. We think that (a) is unlikely, given that TCR activation stimulates Mnk and, in mouse spleen lymphocytes, after CD3 crosslinking, p-eIF4E can be detected (Gorentla et al., 2013). We think that (b) is likely as the primary cells we are dealing with, are from a circulating subset, and consistently other MS studies do not report p-eIF4E in human T cells, example (Howden et al., 2019) (and related dataset). In addition, in classic reports (Kleijn and Proud, 2002) changes of p-eIF4e were not evident upon stimulation of lymphocytes similar to what we used. In the same context, we found that acute Mnk inhibition did not reduce puromycin incorporation in cultured primary human lymphocytes stimulated with anti-CD3/CD28 (Figure 1D). Concerning infiltrating CD3+ cells, the labelling of p-eIF4E by FACS analysis can be gated to a small proportion of cells, 1-2% (Figure 2 —figure supplement 2E). These p-eIF4E positive cells were, interestingly, puromycin positive (same as above) (2). Taken together the data support a model in which eIF4E phosphorylation is important in a specific cellular context.

Given the clear effects of the mTORC1 pathway in vitro, we tested in vivo whether acute Everolimus (mTORC1 blocker) administration affected phosphorylation of rpS6 and puromycin incorporation in infiltrating TILs. Figures 2G-H and Figure 2 —figure supplement 2D show a reduction in both. The results confirm the importance of the mTORC1 pathway in sustaining, in vivo, TILs translation rates.

2) In some parts of the manuscript methodology/analysis should be improved. Specifically, anti-phospho-S240/244 ribosomal protein S6 antibody should be used instead of the antibody recognizing phosphorylated S235/236, as the latter phosphoacceptor sites are not exclusively phosphorylated in a mTOR-dependent manner. In addition, it was found that the statistical tests used were in some cases inappropriate, as illustrated in reviews below.

Several experiments have been repeated with anti-phospho-S240-244. The results are almost overlapping with the ones obtained with anti-phospho-S235-236 (Figures 1C-D; Figure 2—figure supplement 2B-C; Figure 5 —figure supplement 1B). Statistic tests have been checked and amended where necessary (methods).

(3) Assessing co-expression of the markers presented in figures 4D and 5D and relating their levels to protein synthesis activity was thought to be warranted in order to grasp full heterogeneity of tumor-infiltrating T cells.

This point is intriguing. The analysis has been re-done through co-stainings. The results were revealing for CD8^+^ cells, whose relevance in tumor and phenotypic stratification are more clear than for CD4^+^. We found: SLAM6-/TIM3+ cells co-segregating with high puromycin; TIM3+/PD1+ co-segregating with high puromycin (Figures 4E-F); Ki67 and PD-1/TIM3 coexpression (Figure 4 —figure supplement 1). The interpretation is straightforward. Dysfunctional CD8 cells show the highest degree of clonal expansion and proliferation among tumor infiltrating cells (Li, van der Leun, Yofe et al., Cell 2019). Expression of Slamf6 in Tim3+ cells characterizes progenitor exhausted cells that retain polyfunctionality and are responsive to immune checkpoint blockade (ICB) during cancer immunotherapy. Conversely, loss of Slamf6 is associated with terminal exhaustion and loss of responsiveness to ICB, albeit Slamf6-Tim3+ cells maintain high secretion of IFN-γ and proliferative activity (Miller et al., 2019). Our results suggest that these cell subsets are still actively translating and therefore exhaustion is not associated with the loss of translational activity. Accordingly, TIM3+PD-1+ cells exhibited high puromycin incorporation and are an emerging target (Daver et al., 2021). In spite of the semantic confusion these data imply that exhaustion is a continuum and “exhausted” cells are exerting antitumor function, in line with recent works showing that it is the exhausted CD8^+^ T cells that are exerting residual control over tumor growth (Li et al., 2019). See also from line 333, discussion.

Concerning the CD4^+^ cells, the analysis was less meaningful, given their heterogeneity. However, we confirm the contribution of TREGs/CTLA4/TGFα (Shevyrev and Tereshchenko, 2019) to the puromycin pool (Figure 6), which would suggest that immunosuppression is an active cellular activity requiring translation.

4) Finally, it was thought that the manuscript may benefit from additional editing.

Done.

Reviewer #1:In this study, mouse models are employed to systematically establish protein synthesis activity of tumor-infiltrating CD4^+^ and CD8^+^ T lymphocytes in vivo. The effects of stimuli emanating from tumor niche on translational activity of CD4^+^ and CD8^+^ T lymphocytes were also studied. Using ex vivo experiments, the evidence is provided that T cell receptor stimulation induces protein synthesis and mTOR signaling in tumor-infiltrating T cells, whereas hypoxia exerts the opposite effect. Next, the authors used mouse melanoma and colorectal cancer models to confirm these findings. Using the same models, the authors show that tumor-infiltrating CD4^+^ and CD8^+^ T lymphocytes translate more than their counterparts in the spleen. Importantly, it was also observed that tumor-infiltrating lymphocytes also exhibit significant intercellular heterogeneity in translational activity. This was followed by establishing the correlations between levels of protein synthesis and T cell phenotypes using flow cytometry, which revealed strong correlation between levels of protein synthesis and immunophenotypes. Collectively, these findings support the tenet that translation plays a major role in establishing tumor-infiltrating T cell phenotypes, and demonstrate that there is pronounced intracellular heterogeneity in their protein synthesis activity. This heterogeneity in the levels of mRNA translation between tumor-infiltrating T cells may at least in part be explained by the difference in stimuli that these cells are exposed in distinct parts of tumor microenvironment. In long-term, this study may inspire single-cell proteomics and/or similar approaches that do not rely on steady-state mRNA levels to monitor the plasticity of tumor-infiltrating lymphocytes and thus provide molecular bases for more efficient immunotherapies.Strengths: Presented findings further support the critical role of the protein synthesis apparatus in establishing lymphocyte phenotypes, and emphasize the importance of considering alterations that occur downstream of changes in mRNA levels when determining the lymphocyte subtypes in the tumors. This is of particular interest considering recent increase in the number of single cell sequencing studies aiming to address composition of tumor-infiltrating lymphocytes, wherein correlations with phenotypes are driven based on steady-state mRNA composition of the cell. Methodologically, the clear advantage of this study, is that the most of the work was done in vivo, and is therefore of high physiological relevance. In addition, both melanoma and colon cancer models were employed, thereby suggesting that reported observations are likely to be generally applicable, and not cancer type-specific. In addition, the authors monitored the effects of cancer-relevant perturbations on protein synthesis, including T cell receptor stimulation and hypoxia. Overall, this study provides solid evidence that translation plays a major role in establishment of immunophenotypes of the tumor-infiltrating lymphocytes, and that interaction between tumor microenvironment and translational machinery may be responsible for intercellular heterogeneity of T cells in tumor niche.Weaknesses: The major drawback of this study is that it relies on correlation and lacks experiments aiming to decipher mechanisms underlying the interactions between tumor niche and translational machinery of tumor infiltrating lymphocytes. Moreover, it was thought that the focus on mTOR may be too narrow and that other signaling mechanisms known to modulate translation rates in response to environmental cues should also be investigated. Finally, it remains unclear whether the changes in protein synthesis rates affect all the mRNAs uniformly or induce selective perturbations in translational activity of specific subsets of mRNAs.

We acknowledge the mechanistic weakness of this manuscript, given the complexity of dealing with the small numbers and heterogeneity of tumor infiltrating lymphocytes. We analysed eIF2α pathways (Figure 1G; Figure 3E and its supplemental), Mnk-eIF4E (Figure 1D; Figure 2 —figure supplement 2) and manipulated the mTOR pathway with Everolimus (Figure 2H). In this manuscript we address whether (a) high translation in TILs is ubiquitous or selective for some cells, (b) whether it correlates with gene expression changes and (c) is regulated by signalling pathways/environmental clues. It is likely that protein synthesis changes (also) specific mRNA translation given the current body of evidence of how translation works (Hershey et al., 2019). Formal proof, in our work, is lacking. The fact that translating cells have a specific phenotype (Figures 5-6), however, supports the idea of selective mRNA translation.

Notwithstanding these weaknesses, it was thought that with appropriate revision, provided evidence is sufficient to support the major conclusion of the manuscript which is that the interactions between tumor microenvironment and translational machinery play a major role in determining immunophenotypes of tumor-infiltrating lymphocytes and that relying on steady-state mRNA levels may not be sufficient to grasp full plasticity of tumor-infiltrating T cells.– Sole focus of the study is on mTOR signaling. This is somewhat surprising as other mechanisms have been also shown to play a central role in translational reprogramming under hypoxia and T cell receptor stimulation (e.g. ISR, MNK-dependent phosphorylation of eIF4E, etc). Based on this, it was thought that more comprehensive assessment of alterations in signaling pathways that impinge on translational machinery is warranted to pinpoint signaling pathways that mediate the observed effects of tumor microenvironment on T lymphocytes.

The point has been addressed in the general reply (1). We performed an analysis of other pathways, hypoxia-eIF2 and Mnk-eIF4E, and new data have been inserted as detailed in (1) and in the Public Review.

– Conclusions on alterations in mTOR signaling are based solely on monitoring phospho-rpS6 levels. Importantly, the antibody that the authors used recognizes S235/236 on rpS6 which are phosphorylated in cells lacking S6K1 and 2 (PMID: 15060135) and are phosphorylated by other AGC kinases (e.g. RSKs). The phoposho-acceptor site on rpS6 that is exclusively sensitive to alterations in the mTORC1/S6K signaling axis is S240/244 (PMID: 15060135) and thus, antibody recognizing S240/244 should be employed. To strengthen their conclusions, the authors should also add appropriate total antibody controls and monitor other mTORC1 targets (e.g. 4E-BPs, eEF2K).

The point on S240/244 has been addressed in the general reply (2). New data have been inserted in new Figures 1C-D; Figure 2 —figure supplement 2B-C; Figure 5 —figure supplement 1B.

4EBPs: Human and mouse lymphocytes do not express 4E-BP1, but 4E-BP2 (our data; FPKM 4E-BP1: 100-300; FPKM 4E-BP2 3000-5000; (So et al., 2016)). There is an antibody described in the literature that functions in the detection of p-4E-BP (Yi et al., 2017) by FACS. In our hands, it does not give reliable staining in tumor infiltrating lymphocytes (Cell Signalling). Note: the manufacturer shows use of this antibody for the tumor Jurkat T cell line +/- PI3K inhibition, whereas Yi et al., (2017) used it for splenocytes after in vitro stimulation. Both experimental approaches have better cell homogeneity/size/number etc… compared to TILs. In short, it is not that surprising that we fail to see specific staining in infiltrating lymphocytes because they are heterogenous, small, and not stimulated with aCD3/CD28. Similarly, we have not found antibodies for p-eEFK that work effectively by FACS.

– Considering the general lack of mechanistic data to explain heterogeneity in translational activity of tumor-infiltrating T cells, it was thought that monitoring the impact of mTOR inhibitors as well as compounds targeting other signaling mechanisms that adjust protein synthesis in accordance to changes in cellular environment (e.g. ISRIB, MNK inhibitors) on T-cell immunophenotypes and heterogeneity in vivo would be informative.

We tested first in vitro the effects of Mnk inhibitors on global translation, and we did not see differences (Figure 1D). Given the data with mTORC inhibitor Everolimus, we tested its effect in vivo. Since mTORC1 inhibition might have an effect both on tumor cells and T cells, chronic mTORC1 treatment would be confounding. Therefore, we briefly pulsed with Everolimus and assessed the effect on T cells. We found that Everolimus reduces the number of TILs incorporating puromycin, and rpS6 phosphorylation (Figure 2H and its supplemental). ISRIB study will be part of a new project in which we will study eIF2B/eIF2α. We expect that eIF2α phosphorylation is essential for T cell viability and adaptation in vivo, but we have no solid data yet (Rashidi et al., 2020).

– It is well-established that in addition to changes in global protein synthesis levels, stressors such as hypoxia selectively affect translation of different mRNA subsets, whereby the latter changes are likely to significantly contribute to observed immunophenotypes. I understand that translatome-wide translational profiling is out of the scope of this short report and also technically very challenging, but perhaps authors should monitor mRNA and protein levels of selected number of factors that are known to be translationally regulated under these conditions and/or implicated in T cell biology in their system.

We are working on this issue since a long time and it is more complex than what it seems, i.e. since we must score for divergent mRNA/protein expression, and take in account many aspects, mRNA/protein stability, detection threshold, double staining ISH-IHC etc. We are still, unfortunately, (very) far from a decent solution that can be applied to TILs. As this reviewer probably knows, we know some “metabolic” targets regulated at the translational level in cultured CD4^+^ cells, such as ACC1 and PKM (Manfrini et al., 2020; Ricciardi et al., 2018). However, we need to define secreted cytokines, a physiologically relevant endpoint in the tumor microenvironment. IL-2 is one of such (Garcia-Sanz and Lenig, 1996) that may be regulated but we have no formal proof in vivo.

– The article may benefit from some careful editing.

Done.

Reviewer #2:The tumor microenvironment includes the extracellular matrix (ECM) and a multitude of cells including immune cells. During cancer progression, the tumor associated ECM and cell composition/activity is altered. For example, the activity of immune cells towards tumor cells is commonly reduced which facilitates e.g. tumor growth and metastasis. It is well established that, in vitro, activation of T lymphocytes leads to a dramatic upregulation of protein synthesis and increased proliferation. Yet, to what extent this is true in vivo is not yet characterized. In this manuscript, Benedetta De Ponte Conti et. al address this using a FACS-based approach whereby global protein synthesis (quantified by puromycin incorporation) is related to expression of established activation markers for CD4^+^ and CD8^+^ T cells in tumors from the B16/F10 melanoma model. The conclusions are largely in agreement with studies of in vitro models as activation of CD4 or CD8 cells induces proliferate and bolsters protein synthesis. This study thereby pinpoints the close relationship between protein synthesis and immune cell activity also in vivo. One particularly interesting finding is the links between protein synthesis and Treg phenotype. In aggregate, the claims are well supported and important. This study will likely fuel many additional studies trying to in greater detail understand how protein synthesis in immune cells relate to their function within the tumor microenvironment.There are some parts I think could be more developed:1. There is a close link between protein synthesis and proliferation following activation of CD4 and CD8 T cells. This raises the question of how close puromycin incorporation associates with proliferation. In figure 5 S1, Ki67 vs protein synthesis is assessed in CD4^+^ T cells. This suggests a very close relationship. IS there a corresponding relationship for CD8^+^ T cells? Are there any aspects of the protein synthesis profile which do not associate with proliferation? What would the phenotype of such cells be?

We performed the analysis of Ki67 in CD8^+^ T cells and found a positive correlation with proliferation (new Figure 4C). In addition, we found a positive relationship in CD8^+^ cells between PD-1 expression and Ki-67 labelling, as well as with TIM3 and Ki67 (new Figure 4C and Figure 4 —figure supplement 1). The markers that we identified (Figure 4), in the context of CD8^+^ tumor infiltrating lymphocytes can be compared to the ones found in other studies. It is evident that clonal expansion, as detected, by TCR sequencing (Liu et al., 2020; Zhang et al., 2018) is a major phenomenon that coincides with our observations on protein synthesis (discussion, line 320).

CD4^+^ helper cells are highly heterogenous and a similar conclusion, e.g. whether high translation coincides with clonal expansion, is more difficult. However, Puro+ Tregs far exceed Puro + Tconv (Figure 6D) and, together with the relatively limited EOMES+ Tr1 subset, constituting 1/20 of the FOXP3+ repertoire, they are highly expanded in vivo (Bonnal et al., 2021). In conclusion, highly translating CD4^+^ cells are mostly associated with clonal expansion of Tregs.

Thus, clonal expansion driven by TCR stimulation is a major source of translating TILs.

2. All analyses are performed on populations of cells but there are no cross-comparisons between marker expression. I.e. is there one subset of T cells with high protein synthesis which express most of these markers (Figure 4D) or are there subsets of high protein synthesis cells which express subsets of markers? I am not sure if it is technical possible, but it would be interesting to assess co-expression of the identified markers in figure 4D and 5D and how their expression relates to protein synthesis.

Multiple labelling was performed. We repeated stainings and analysis, maintaining a rigid gating. This allows the definition of CD8^+^ TIM3+PD1+ and CD8^+^, TIM3+, SLAMF6- subsets described in point 3. Data indicate also CD4^+^, FOXP3+, CTLA4+, as highly translating (Figures 5 and 6).

3. Analysis is largely qualitative when assessing expression of makers and protein synthesis. Perhaps it would be informative to also perform a quantitative analysis within single tumors?

Perhaps, we missed the point. The analysis has been quantitative in single tumors. Example: Figure 2, Supp. 2B, right, each dot represents the percentage of cells positive for pS6/Puro+ and pS6/Puro- in a given tumor. The 2D-plot on the left is the representative distribution of density in a single tumor. Data can be alternatively represented as in Figure 3D, where the trajectory of single individuals can be seen, although we thought that this was reasonable only for PMO distribution, since it is a spatial parameter in the tissue (histological). I hope we have clarified. Rephrasing: Figure 2, Figure supplement 2B, right derives from 40 single stainings, same for 2C, etc.

4. It is unclear to me what the relationship between T cell exhaustion and protein synthesis (and also proliferation) is based on the presented data. It seems logical to expect reduced protein synthesis in exhausted T cells, but this may not be the case.

Please see point (3) of mandatory changes. We inserted a discussion statement. Exhaustion is not associated with the loss of translational activity. The topic is in itself confusing because exhaustion may be a continuum or poorly defined. Exhausted CD8^+^ T cells still play a critical role in cancer. TILs expressing markers of exhaustion (PD-1, LAG3 and TIM3) are more likely to express IFN-γ(Gros et al., 2014). While exhausted CD8^+^ T cells have reduced function as compared to those elicited by an acute infection, several data suggest that it is the exhausted CD8^+^ T cells that are exerting residual control over tumor growth (Li et al., 2019). In short, in vivo, exhaustion is a developmental stage that can be reactivated. Unfortunately, the terminology is confusing.

Reviewer #3:Overall the figures and the text were clear and concise and the data well described. The data shown support the claims made, but the authors have not examined the other major pathway for regulation of translation. My main issues relate to the first three figures.1. The authors focus on the regulation of protein synthesis through mTOR, however translation initiation is also regulated by eIF2. Moreover, the PERK-eIF2-ATF4 axis has been shown to be important for tumour cell adaptation to stress – therefore, as all four eIF2 kinases are expressed in these models, it is important to determine any effect from eIF2 on translation activation/inhibition:– What are the levels of eIF2α phosphorylation in the -/+puromycin cell populations (Figure 1 and 2)?– As hypoxia is known to regulate eIF2 kinases what is the activation status of the four kinases in these model systems during hypoxia (Figure 1 – 3)?

Please see also the general response to mandatory corrections. We have now demonstrated by ELISA eIF2α phosphorylation in hypoxic, lowly translating CD4^+^ cells (new Figure 1G). in vivo, we can see that hypoxia-rich regions contain CD4^+^/peIF2a+ cells (new Figure 3 and its supplemental). We call them peIF2a+ immunoreactive-like cells. We cannot perform parallel eIF2 total staining in immunohistochemistry (antibodies of same species). Available Anti-phospho-eIF2α antibodies were not good for FACS analysis in TILs.

The point on the eIF2α kinases is intriguing. Our RNAseq data on human TILs show that all 4 kinases are expressed (Bonnal et al., 2021), with PKR and HRI predominant, and gcn2, PERK 5-fold less. For protein validation, concerning the four kinases, issues of antibody quality and FACS suitability are even worse than for p-eIF2α. Phospho HRK to my knowledge has never been produced. Phospho-PKR by CST has been discontinued. Phospho-GCN2 worked only in large cells. We agree that PERK axis is probably the more relevant. According to the literature, hypoxia leads to phosphorylation of eIF2α by PERK, for instance (Koumenis et al., 2002). In addition, in vivo, hypoxia is accompanied by AMPK activation, which in turn has a feedback on translation (Horman et al., 2002). In short, we think that this issue must be (and will be) treated as a new project on eIF2 kinases (see discussion).

– It is known that mTOR signalling can crosstalk to eIF2 signalling, and vice-versa, so how do the levels of eIF2α phosphorylation (and activity of eIF2 kinases) correlate with mTOR activity in these models during hypoxia (Figure 1 – 3)?– To support the author's conclusions it would be necessary to show that the regulation of translation in this model is independent of eIF2 signalling.

We agree. It is known that mTOR crosstalks with eIF2 and the extent of potential cross-talk of signalling is huge (Roux and Topisirovic, 2018). However, please note that we never concluded that translation is independent of eIF2 signalling, but we stated that hypoxia in combination with TCR stimulation were the main regulators of the TILs rate of translation, and high translation was associated with specific TIL phenotypes. We hope that this is now clear and that the new data on eIF2a that were added, at least partly, contribute to improving the issue.

2. The author's state at various points that the data shows mTOR activity is increased in the puromycin positive proportion of cells, however they do not directly show this:– The antibody used for phospho-RPS6 recognises residues (Ser235/236), which can be phosphorylated in the absence of S6K1 and S6K2, therefore, could this regulation be independent of mTOR? Although RPS6 is downstream of mTOR signalling it is not a direct substrate of mTOR, therefore, to fully support these conclusions, blots for direct substrates of mTOR (p-S6K1, p-4EBP1), with corresponding blot for all total proteins, should be shown.

We agree with the comment. We have inserted data on S240/244 which is dependent on S6K1/2 and we found that it behaves similarly to S235/236 (Figures 1C-D; Figure 2 —figure supplement 2B-C; Figure 5 —figure supplement 1B). The advantage of detecting S235/236 rpS6 phosphorylation was its outstanding performance by FACS.

4EBPs: Human and mouse lymphocytes do not express 4E-BP1, but 4E-BP2 (our data; FPKM 4E-BP1: 100-300; FPKM 4E-BP2 3000-5000; (So et al., 2016)). There is an antibody described in the literature that functions in the detection of p-4E-BP (Yi et al., 2017) by FACS. In our hands, it does not give reliable staining in tumor infiltrating lymphocytes.

3. Figure 1C – Puromycin incorporation demonstrates that protein synthesis is increased yet only the phospho-RPS6 blot is included. A total RPS6 blot must be included to demonstrate that the increase in phospho-RPS6 signal is not simply increased synthesis of total RPS6. This is especially important as the abundance of vinculin (used as the loading control) varies across samples in Figure 1B and 1C.

This is completely true and biologically relevant. Total S6 has been added. In line with the literature also rpS6 protein levels increase with stimulation (Howden et al., 2019), which is not surprising since they are TOP mRNAs (Loreni et al., 2014). It is the total absence of phospho-rpS6 in unstimulated cells the real feature of quiescent human T cells. Total S6 does not work in FACS.

4. Figure 1F – statistical analysis – multiple comparisons have been made to the 0 hr time point. Running multiple t-tests in this manner increases the chance of error, therefore, statistical analysis should take these multiple comparisons into account by using ANOVA with Dunnett's post hoc test.

We now performed the suggested tests with similar results.

References

Bonnal, R.J.P., Rossetti, G., Lugli, E., De Simone, M., Gruarin, P., Brummelman, J., Drufuca, L., Passaro, M., Bason, R., Gervasoni, F.*, et al.* (2021). Clonally expanded EOMES(+) Tr1-like cells in primary and metastatic tumors are associated with disease progression. Nat Immunol *22*, 735-745.

Daver, N., Alotaibi, A.S., Bucklein, V., and Subklewe, M. (2021). T-cell-based immunotherapy of acute myeloid leukemia: current concepts and future developments. Leukemia *35*, 1843-1863.

Garcia-Sanz, J.A., and Lenig, D. (1996). Translational control of interleukin 2 messenger RNA as a molecular mechanism of T cell anergy. J Exp Med *184*, 159-164.

Gorentla, B.K., Krishna, S., Shin, J., Inoue, M., Shinohara, M.L., Grayson, J.M., Fukunaga, R., and Zhong, X.P. (2013). Mnk1 and 2 are dispensable for T cell development and activation but important for the pathogenesis of experimental autoimmune encephalomyelitis. J Immunol *190*, 1026-1037.

Gros, A., Robbins, P.F., Yao, X., Li, Y.F., Turcotte, S., Tran, E., Wunderlich, J.R., Mixon, A., Farid, S., Dudley, M.E.*, et al.* (2014). PD-1 identifies the patient-specific CD8(+) tumor-reactive repertoire infiltrating human tumors. J Clin Invest *124*, 2246-2259.

Hershey, J.W.B., Sonenberg, N., and Mathews, M.B. (2019). Principles of Translational Control. Cold Spring Harb Perspect Biol *11*.

Horman, S., Browne, G., Krause, U., Patel, J., Vertommen, D., Bertrand, L., Lavoinne, A., Hue, L., Proud, C., and Rider, M. (2002). Activation of AMP-activated protein kinase leads to the phosphorylation of elongation factor 2 and an inhibition of protein synthesis. Curr Biol *12*, 1419-1423.

Howden, A.J.M., Hukelmann, J.L., Brenes, A., Spinelli, L., Sinclair, L.V., Lamond, A.I., and Cantrell, D.A. (2019). Quantitative analysis of T cell proteomes and environmental sensors during T cell differentiation. Nat Immunol *20*, 1542-1554.

Kleijn, M., and Proud, C.G. (2002). The regulation of protein synthesis and translation factors by CD3 and CD28 in human primary T lymphocytes. BMC Biochem *3*, 11.

Koumenis, C., Naczki, C., Koritzinsky, M., Rastani, S., Diehl, A., Sonenberg, N., Koromilas, A., and Wouters, B.G. (2002). Regulation of protein synthesis by hypoxia via activation of the endoplasmic reticulum kinase PERK and phosphorylation of the translation initiation factor eIF2alpha. Mol Cell Biol *22*, 7405-7416.

Li, H., van der Leun, A.M., Yofe, I., Lubling, Y., Gelbard-Solodkin, D., van Akkooi, A.C.J., van den Braber, M., Rozeman, E.A., Haanen, J., Blank, C.U.*, et al.* (2019). Dysfunctional CD8 T Cells Form a Proliferative, Dynamically Regulated Compartment within Human Melanoma. Cell *176*, 775-789 e718.

Liu, Z., Li, J.P., Chen, M., Wu, M., Shi, Y., Li, W., Teijaro, J.R., and Wu, P. (2020). Detecting Tumor Antigen-Specific T Cells via Interaction-Dependent Fucosyl-Biotinylation. Cell *183*, 1117-1133 e1119.

Loreni, F., Mancino, M., and Biffo, S. (2014). Translation factors and ribosomal proteins control tumor onset and progression: how? Oncogene *33*, 2145-2156.

Ma, C.Y., Marioni, J.C., Griffiths, G.M., and Richard, A.C. (2020). Stimulation strength controls the rate of initiation but not the molecular organisation of TCR-induced signalling. *eLife 9*.

Manfrini, N., Ricciardi, S., Alfieri, R., Ventura, G., Calamita, P., Favalli, A., and Biffo, S. (2020). Ribosome profiling unveils translational regulation of metabolic enzymes in primary CD4(+) Th1 cells. Dev Comp Immunol *109*, 103697.

Miller, B.C., Sen, D.R., Al Abosy, R., Bi, K., Virkud, Y.V., LaFleur, M.W., Yates, K.B., Lako, A., Felt, K., Naik, G.S.*, et al.* (2019). Subsets of exhausted CD8(+) T cells differentially mediate tumor control and respond to checkpoint blockade. Nat Immunol *20*, 326-336.

Rashidi, A., Miska, J., Lee-Chang, C., Kanojia, D., Panek, W.K., Lopez-Rosas, A., Zhang, P., Han, Y., Xiao, T., Pituch, K.C.*, et al.* (2020). GCN2 is essential for CD8(+) T cell survival and function in murine models of malignant glioma. Cancer Immunol Immunother *69*, 81-94.

Ricciardi, S., Manfrini, N., Alfieri, R., Calamita, P., Crosti, M.C., Gallo, S., Muller, R., Pagani, M., Abrignani, S., and Biffo, S. (2018). The Translational Machinery of Human CD4(+) T Cells Is Poised for Activation and Controls the Switch from Quiescence to Metabolic Remodeling. Cell Metab *28*, 895-906 e895.

Roux, P.P., and Topisirovic, I. (2018). Signaling Pathways Involved in the Regulation of mRNA Translation. Mol Cell Biol *38*.

Shevyrev, D., and Tereshchenko, V. (2019). Treg Heterogeneity, Function, and Homeostasis. Front Immunol *10*, 3100.

So, L., Lee, J., Palafox, M., Mallya, S., Woxland, C.G., Arguello, M., Truitt, M.L., Sonenberg, N., Ruggero, D., and Fruman, D.A. (2016). The 4E-BP-eIF4E axis promotes rapamycin-sensitive growth and proliferation in lymphocytes. Sci Signal *9*, ra57.

Yi, W., Gupta, S., Ricker, E., Manni, M., Jessberger, R., Chinenov, Y., Molina, H., and Pernis, A.B. (2017). The mTORC1-4E-BP-eIF4E axis controls de novo Bcl6 protein synthesis in T cells and systemic autoimmunity. Nat Commun *8*, 254.

Zhang, L., Yu, X., Zheng, L., Zhang, Y., Li, Y., Fang, Q., Gao, R., Kang, B., Zhang, Q., Huang, J.Y.*, et al.* (2018). Lineage tracking reveals dynamic relationships of T cells in colorectal cancer. Nature *564*, 268-272.